# A common genetic mechanism underlies morphological diversity in fruits and other plant organs

Shan Wu [1,13], Biyao Zhang[2], Neda Keyhaninejad[1,2], Gustavo R. Rodríguez [1,3], Hyun Jung Kim[1,14], Manohar Chakrabarti[1,15], Eudald Illa-Berenguer [2], Nathan K. Taitano[2], M. J Gonzalo[4], Aurora Díaz[4,5], Yupeng Pan[6], Courtney P. Leisner [7], Dennis Halterman[8], C. Robin Buell[7], Yiqun Weng[8], Shelley H. Jansky[8], Herman van Eck[9], Johan Willemsen[9], Antonio J. Monforte [4], Tea Meulia[10,11] & Esther van der Knaap[1,2,12]

Shapes of edible plant organs vary dramatically among and within crop plants. To explain and ultimately employ this variation towards crop improvement, we determined the genetic, molecular and cellular bases of fruit shape diversity in tomato. Through positional cloning, protein interaction studies, and genome editing, we report that OVATE Family Proteins and TONNEAU1 Recruiting Motif proteins regulate cell division patterns in ovary development to alter final fruit shape. The physical interactions between the members of these two families are necessary for dynamic relocalization of the protein complexes to different cellular compartments when expressed in tobacco leaf cells. Together with data from other domesticated crops and model plant species, the protein interaction studies provide possible mechanistic insights into the regulation of morphological variation in plants and a framework that may apply to organ growth in all plant species.

[1] Department of Horticulture and Crop Science, The Ohio State University, 1680 Madison Ave, Wooster, OH 44691, USA. [2] Institute of Plant Breeding, Genetics and Genomics, University of Georgia, 111 Riverbend Rd, Athens, GA 30602, USA. [3] Instituto de Investigaciones en Ciencias Agrarias de Rosario, Cátedra de Genética Facultad de Ciencias Agrarias UNR, Campo Experimental Villarino, S2125ZAA Zavalla, Santa Fe, Argentina. [4] Instituto de Biología Molecular y Celular de Plantas, Universitat Politècnica de València-Consejo Superior de Investigaciones Científicas, Ingeniero Fausto Elio s/n, 46022 Valencia, Spain. [5] Unidad de Hortofruticultura, Instituto Agroalimentario de Aragón, CITA-Universidad de Zaragoza, Avenida de Montañana 930, 50059 Zaragoza, Spain. [6] Horticulture Department, University of Wisconsin, Madison, WI 53706, USA. [7] Plant Biology Department, Michigan State University, East Lansing, MI 48824, USA. [8] Vegetable Crops Research Unit, USDA-ARS, 1575 Linden Drive, Madison, WI 53706, USA. [9] Plant Breeding, Wageningen University and Research, P.O.Box 3866700 AJ Wageningen, The Netherlands. [10] Molecular and Cellular Imaging Center, The Ohio State University, 1680 Madison Ave, Wooster, OH 44691, USA. [11] Department of Plant Pathology, The Ohio State University, 1680 Madison Ave, Wooster, OH 44691, USA. [12] Department of Horticulture, University of Georgia, 3111 Carlton Rd, Athens, GA 30602, USA. [13]Present address: Boyce Thompson Institute, 533 Tower Rd, Ithaca, NY 14853, USA. [14]Present address: Division of SMART Horticulture, Yonam College, Cheonan 31005, South Korea. [15]Present address: Department of Plant and Soil Sciences, University of Kentucky, Lexington, KY 40506, USA. Correspondence and requests for materials should be addressed to E.Knaap. (email: vanderkn@uga.edu)

Remarkable phenotypic diversity characterizes cultivated fruits and vegetables. Thousands of years of selection under cultivation have allowed for the accumulation of mutations that collectively comprise modern day germplasm. This diversity is critical for the successful marketing of a wide array of plant species, from food crops to ornamentals. Utilizing morphologically diverse domesticated germplasm also leads to a better understanding of fundamental growth processes common to all higher plants. Coordinated cell division and expansion patterns regulate the shape and size of plant organs. Specifically, the rate, duration and plane of cell division, as well as isotropic and anisotropic cell enlargement contribute greatly to final morphology of plant organs[1]. Recent studies have revealed several genes that control the growth form of agriculturally important organs, such as fruits and seeds[2,3]. One of the most commonly utilized tomato fruit shape gene is *OVATE*[4], the founding member of the *OVATE Family Protein* (OFP) class[5]. A mutation in *OVATE* usually leads to an elongated fruit, yet the extent of elongation varies depending on the genetic background[6]. Shape regulation of the aerial parts in Arabidopsis and grain shape in rice is controlled in part by *LONGIFOLIA1* and *LONGIFOLIA2*[7–10], members of the *TONNEAU1 Recruiting Motif* (*TRM*) family[11]. Whether *OFPs* and *TRMs* function in the same developmental pathway to regulate organ shape and whether orthologous members in other plants share similar functions is unknown. Moreover, the biochemical function and the impacts of the encoded proteins at the cellular and tissue-specific levels are poorly understood. Thus, gaining insights into the function of these proteins in developmental processes related to shape determination will be key to understanding the remarkable morphological diversity observed in plant organs within and among species.

Herein we fine-map and clone a *suppressor of ovate* (*sov1*), which encodes another member of the OFP class. The altered shape is due to changes in cell number in the ovary, which implies a shift in cell division patterning during development of the organ. OVATE-interacting proteins, identified by the yeast two-hybrid (Y2H) approach, include 11 members of the tomato TRM family. We show that conserved charged amino acid residues in the interacting motifs are necessary for the protein-protein interaction. Upon co-expression in *Nicotiana benthamiana* epidermal leaf cells, the protein complexes relocalize subcellularly to the cytosol or to microtubules depending on the OFP and TRM combination. Using CRISPR, a knock out mutation in tomato *TRM5* rescues the fruit shape phenotype in the *ovate/sov1* double mutants. We also demonstrate that members of the same *OFP* and *TRM* subclades are associated with natural variation of melon and cucumber fruit as well as potato tuber shape. These results integrate the previous knowledge on OFPs and TRMs, and provide possible mechanistic insights into organ shape regulation in higher plants.

## Results

### Fine-mapping and cloning of *SlOFP20*.
The mutant allele of *ovate* is common in cultivated tomato and usually leads to an elongated fruit shape[6]. Tomato accessions carrying *ovate* may produce oval, rectangular or pear shaped fruit, and a few produce fruit with a pointed tip. In rare cases, *ovate* accessions produce fruits of a near perfect round shape[6]. These variable tomato fruit shapes suggest the presence of modifier loci in the germplasm, and knowledge of the underlying genes would provide insights into the regulation of morphological diversity. As a step toward identifying critical genes in the *OVATE* fruit shape pathway, we mapped one modifier locus on chromosome 10 that we refer to as *suppressor of ovate* (*sov1*)[12]. Further fine-mapping of the locus identified a 149.7-kb region (Supplementary Table 1; Fig. 1a) that carries three annotated genes, *Solyc10g076170*, *Solyc10g076180* and *Solyc10g076190*. The gene *Solyc10g076180* corresponds to another member of the OFP class, *SlOFP20*, which was highly expressed in flowers at anthesis (Supplementary Fig. 1a). Of these three genes, only *SlOFP20* was consistently differentially expressed in near isogenic lines (NIL) polymorphic at the *sov1* locus (Fig. 1b), suggesting that this gene might control fruit shape. Investigations into allelic diversity at *sov1* showed a 31-kb deletion in the upstream regulatory region of *SlOFP20* 6.5 kb away from the transcription start site, which may be the cause of the reduced expression of the gene (Fig. 1a; Supplementary Fig. 2). The other candidate genes were either not differentially or extremely low expressed, or encoded for an unknown protein, and therefore, were not pursued further.

To confirm that *SlOFP20* is indeed underlying *sov1*, we overexpressed the gene in the Yellow Pear variety (*Solanum lycopersicum* L.) (Fig. 1c). Conversely, we reduced expression of the wild type *SlOFP20* allele in the close wild relative of tomato, *S. pimpinellifolium* LA1589 to approximate the expression level in the NIL (Fig. 1b). The results showed that overexpression of *SlOFP20* in Yellow Pear produced much rounder fruits than the non-transformed Yellow Pear as well as shortened leaves and rounder leaflets (Fig. 1c; Supplementary Fig. 3; Supplementary Data 1). Correspondingly, down regulation of *SlOFP20* in LA1589 led to a more elongated fruit shape but only in the *ovate* background (Supplementary Data 2). Importantly, down regulation of *SlOFP20* in the *ovate* background led to a tomato shape that was as elongated as or more elongated than the natural *sov1/ovate* double NIL (Fig. 1d). These plant transformation results conclusively show that *SlOFP20* underlies *sov1* and demonstrate that mutations in two OFP members contribute to natural fruit shape variation in the tomato germplasm.

### OFPs and TRMs relocalize upon interaction.
To determine the role of OFPs in fruit morphology, we identified interacting proteins using the Y2H approach. Using OVATE as the bait, we identified 185 interacting clones, of which 63.8% originated from 11 genes belonging to the TRM superfamily (Fig. 2a; Supplementary Table 2). Phylogenetic analysis showed that several subgroups represented OVATE-interacting tomato TRMs (Supplementary Fig. 4). TRMs function in assembling the TTP (TON1-TRM-PP2A) complex, and certain TRMs target this complex to microtubules[11]. The TTP complex is postulated to regulate the organization of microtubule arrays, and thus control cell division patterns and cell growth[11,13,14]. Based on the minimum interacting protein region, motif M8 was found in all OVATE-interacting SlTRMs, and conversely, SlTRMs without a predicted M8 motif were not recovered. This finding suggests that M8 is required for interaction with OVATE (Supplementary Table 3). Pairwise Y2H interactions of SlOFP20 and selected SlTRMs showed that several interacted with both OFPs, presumably through the conserved C-terminal OFP domain (Fig. 2d).

To test whether the negatively charged residues in the OFP domain interact with the positively charged residues in the M8 motif of the OVATE-interacting TRMs, we performed Y2H assays with wild-type and mutant versions of the proteins (Fig. 2d; Supplementary Fig. 5). In general, mutations in conserved acidic residues in the OFP domain led to reduced or abolished Y2H interaction with different TRMs, whereas mutations in less conserved acidic residues did not change interactions in yeast with SlTRM compared to the wild-type allele (Fig. 2b, d; Supplementary Fig. 5). Conversely, mutations in the basic residue of the SlTRM M8 motif reduced the interactions with wild-type OVATE and SlOFP20 (Fig. 2c–d). Mutations of D280 in OVATE

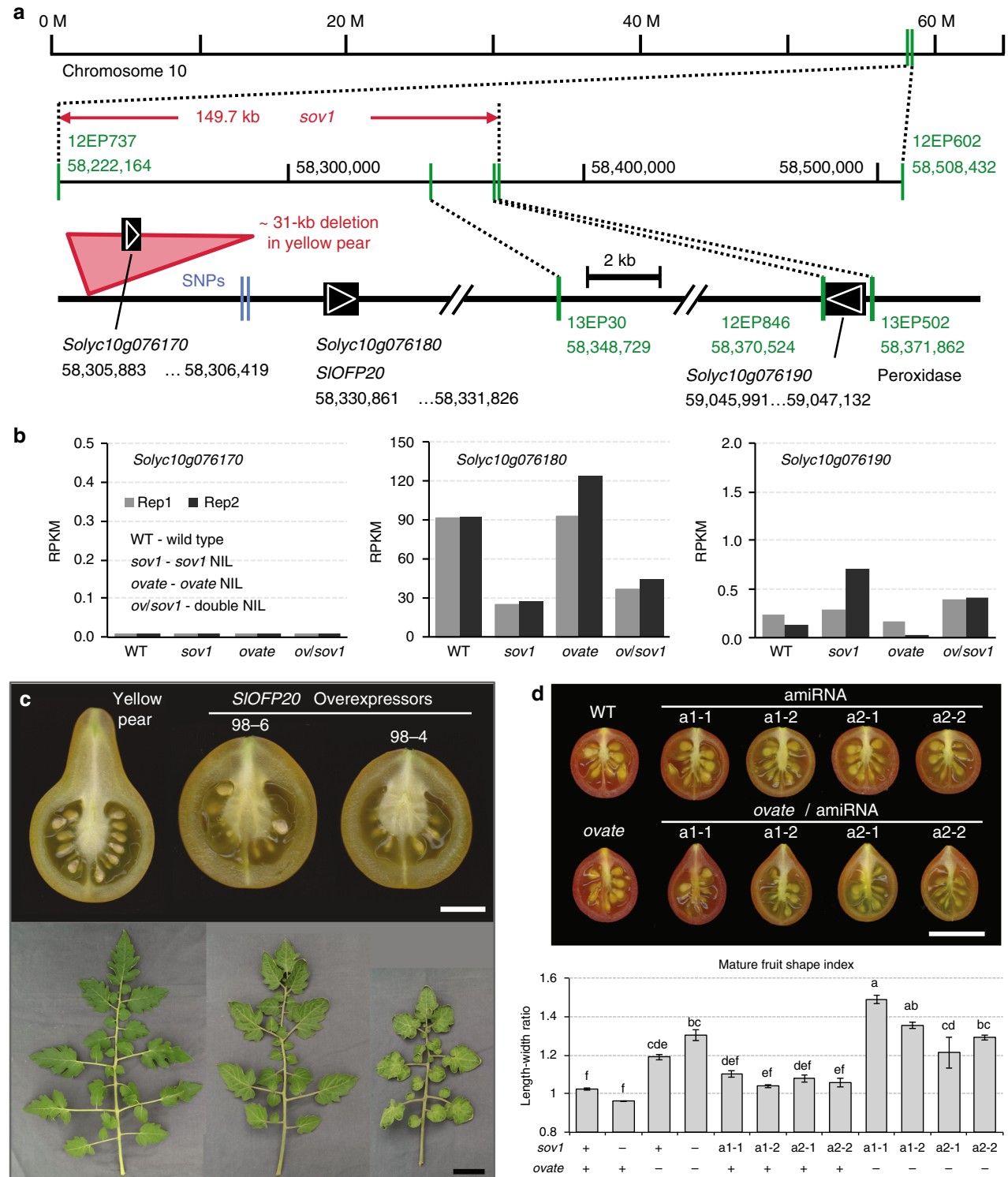

**Fig. 1** Map-based cloning of *SlOFP20*. **a** The 149.7-kb *sov1* locus on tomato chromosome 10. Markers are indicated in green. The physical positions in base pairs are labeled beneath markers and genes. **b** Expression levels of the three genes within the *sov1* locus in anthesis-stage tomato flowers from the *ovate* and *sov1* NILs. RPKM, reads per kilobase of transcript per million mapped reads. **c** Effect of overexpression of *SlOFP20* on fruits and leaves of Yellow Pear in two independent $T_O$ transgenic lines. **d** Effect of down regulation of *SlOFP20*. The lines are $BC_1F_2$ from four independently transformed lines expressing two different artificial microRNAs (amiRNAs, a1 and a2) targeting *SlOFP20*. The error bars and letters in the graph indicate the standard errors among four plants and significant differences in fruit shape index evaluated by Tukey's test ($\alpha < 0.05$), respectively. +, wild type allele; -, mutant allele. The white and black bars represent 1 cm and 5 cm, respectively

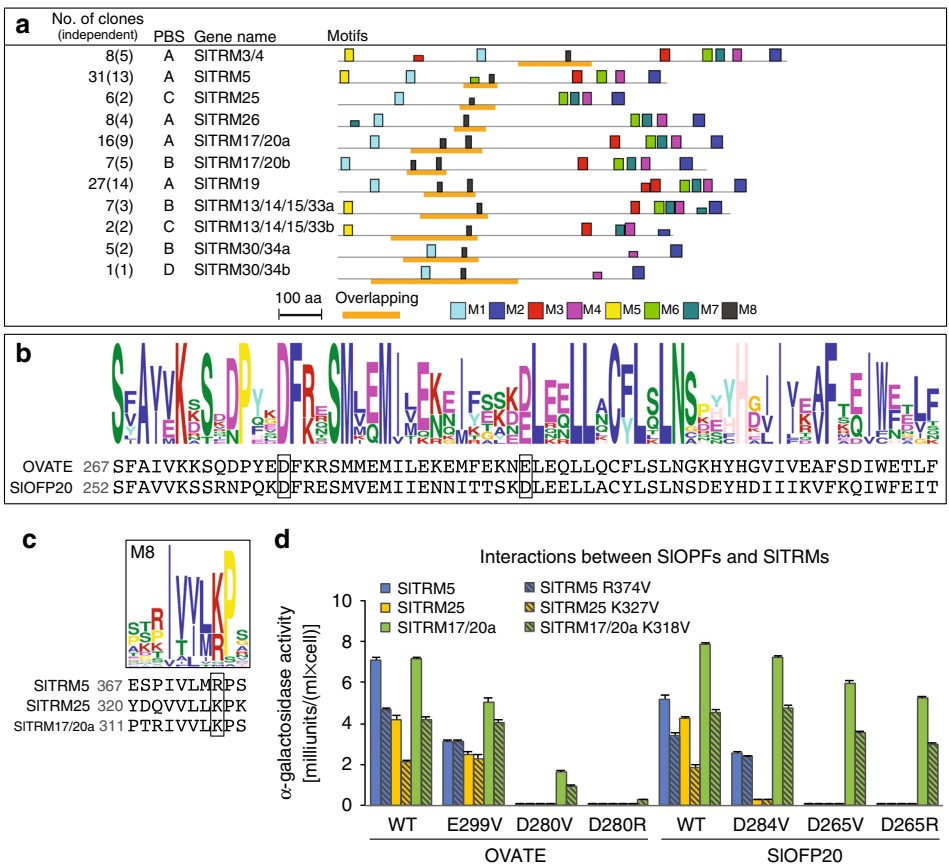

**Fig. 2** Interactions of SlOFPs and SlTRMs in yeast. **a** The 11 SlTRMs identified in the Y2H screen to interact with OVATE. The Predicted Biological Score (PBS) A–D represents the highest to lowest interaction confidence scores, respectively. The eight motifs are indicated by the colored boxes. The orange bar highlights the overlapping region of independent *SlTRM* clones that were identified in the screen. **b** Conserved amino acids in the OFP domain. The boxed regions indicate the in vitro mutagenized amino acids. **c** The consensus TRM M8 motif. **d** Interactions between wild-type and mutant SlOFPs and SlTRMs in yeast. The error bars indicate the standard errors among three colonies

and the equivalent D265 in SlOFP20 most severely reduced or abolished the interactions with the TRMs. The responses of TRMs to OFP mutations differed, with SlTRM17/20a being less affected than SlTRM5, demonstrating in yeast the potentially dissimilar kinetics between the members of these two families. Together, these data validate the Y2H interactions and imply that the D280 residue in OVATE (D265 in SlOFP20) and the M8 K/R residues are most critical for protein-protein interactions between OFPs and TRMs.

To determine whether OVATE and SlOFP20 physically interact with SlTRMs in plant cells as they do in yeast, we expressed the proteins alone or together in pairs in tobacco (*N. benthamiana*) leaf epidermal cells. Many OFPs localize to the nucleus[5,15], whereas TRMs localize in different subcellular locations including the cytoskeleton and cytoplasm[11]. The subcellular localization of OVATE and SlOFP20 was distinct as the former localized to the cytoplasm and the latter to the nucleus and cytoplasm (Fig. 3a–b). For the TRMs, we evaluated the subcellular localization of SlTRM5 and SlTRM17/20a, which are strong interactors with OVATE in yeast (Fig. 2d). When expressed alone, SlTRM5 associated with the cytoskeleton (Fig. 3c) whereas SlTRM17/20a was in the cytoplasm (Fig. 3d). The cytoskeleton interaction of SlTRM5 was most likely with microtubules as we observed a similarity in patterning with microtubule markers and the disruption of SlTRM5 subcellular localization in the presence of the microtubule-destabilizing agent Oryzalin (Supplementary Fig. 6). Henceforth, we consider the cytoskeleton-localized SlTRM5 to be a microtubule-associated

protein. Regardless of the localization of the two OFP members, neither localized in the same compartment as SlTRM5 when expressed alone. To test the hypothesis that these proteins may relocate upon interaction, the OFPs and SlTRM5 were co-expressed in tobacco leaf cells. The results showed that SlTRM5 re-localized nearly exclusively to the cytoplasm when co-expressed with OVATE while the latter remained in the cytoplasm (Fig. 3e; Supplementary Fig. 7a). On the other hand, when co-expressed with SlOFP20, SlTRM5 remained associated with the microtubules in 71% of the cells (Fig. 3f; Supplementary Fig. 7a). Distinct from OVATE's localization (Fig. 3e; Supplementary Fig. 7b), co-expression with SlTRM5 led to relocalization of SlOFP20 to the microtubules (Fig. 3f; Supplementary Fig. 7c). Mutants of OVATE, OFP20, and SlTRM5 that led to reduced or abolished interaction in yeast also greatly reduced the number of cells that showed relocalization of SlTRM5 when co-expressed with OVATE[D280R] or SlOFP20[D265R], and double mutants reduced relocalization of either SlOFP20 or SlTRM5 even further (Supplementary Fig. 7d-e). Similar results were obtained with another microtubule-associated TRM, SlTRM3/4 (Supplementary Fig. 8). When co-expressed with either OVATE or SlOFP20, SlTRM3/4 re-localized entirely to the cytoplasm, and similar decreases in relocalization were observed with mutant versions of OVATE (OVATE[D280R]) and SlTRM3/4 (SlTRM3/4[K560V]). Bifluorescence Complementation (BiFC) assays further supported the interactions of OVATE and SlOFP20 with several TRMs (Fig. 3g–i; Supplementary Fig. 8g) including the subcellular location of these interactions in the cytosol or microtubules.

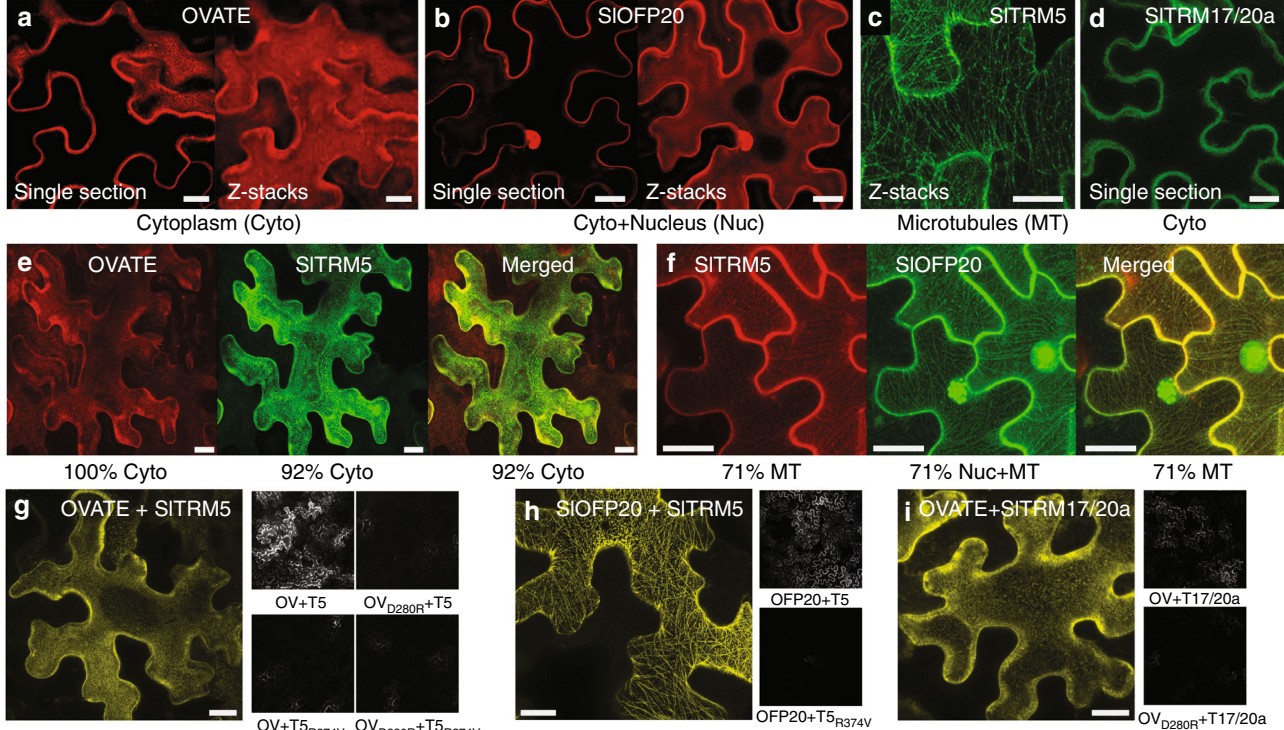

**Fig. 3** Interactions of SlOFPs and SlTRMs in *N. benthamiana* leaf epidermal cells. **a–d** Subcellular localization of OVATE, SlOFP20, SlTRM5, SlTRM17/20a, respectively. **e** Co-expression of OVATE and SlTRM5. **f** Co-expression SlOFP20 and SlTRM5. **g–i** BiFC with wild-type and mutant versions of SlOFPs and SlTRMs. On the left, an image of a single cell at 40× magnification demonstrating the subcellular localization of the interacting proteins. Black and white images on the right, an overview of signal intensity at 10× magnification. Scale bar, 20 μm

Consistent with the co-localization experiments, the BiFC signal in the tobacco leaf cells was substantially reduced when co-expressing mutant versions of the proteins. In all, these results demonstrate that both SlOFPs and SlTRMs physically interact in plant cells and that this contact leads to relocalization of one or the other protein. Moreover, the interaction between specific OFPs and TRMs appears dynamic because of different relocalization patterns for each pair in the tobacco leaf epidermal cell system.

**trm5 rescues ovate/sov1 by changing cell division patterns**. To investigate the effect of the two tomato OFPs with SlTRMs on fruit shape in a common genetic background, we constructed NILs for the natural *ovate* and *sov1* alleles and created mutations in the tomato TRMs that represent the TRM1–5 clade in Arabidopsis (Supplementary Fig. 4 and 9). The OVATE-interacting TRMs, *SlTRM3/4* and *SlTRM5* were the only members of this clade and, thus were pursued further. Moreover, gene expression analyses showed that among others, *SlTRM5* was highly co-expressed with *OVATE* (Supplementary Fig. 1b). The natural *sov1* NIL, carrying the mutation in the *SlOFP20* promoter, produced fruit that was morphologically indistinguishable from wild type (Fig. 4a–b). The *ovate* NIL showed an increase in fruit elongation and degree of obovoid, and a decrease in proximal end angle, whereas the double NIL showed the strongest effect on fruit shape. Both mutant alleles were recessive and studies of epistasis showed a strong synergistic interaction between the two loci (Supplementary Table 4). Whereas the inactivation of *SlTRM3/4* led to inconclusive results (Supplementary Fig. 10), the inactivation of *SlTRM5* resulted in a slightly flatter fruit. Backcrossing *Sltrm5–1* into the *ovate/sov1* double NIL background resulted in greatly reduced fruit elongation, with shapes that were similar to

wild-type fruits (Fig. 4a, b; Supplementary Table 5). Another inactivation allele, *Sltrm5–2*, showed similar results as *Sltrm5–1* (Supplementary Fig. 11). Therefore, *Sltrm5* complements fruit shape in the *ovate/sov1* background.

To determine whether these mutations exert their effects during either ovary or fruit development, we investigated ovary shape at flower opening. All mutants, including *sov1* and *Sltrm5*, showed significant ovary shape changes at anthesis (Fig. 4c; Supplementary Table 5). The most significant morphological changes appeared at the proximal end of the ovary, away from the ovules. This area expanded strongly in the proximo-distal direction in the *ovate* and *ovate/sov1* mutants and returned to the wild-type phenotype in the *ovate/sov1/Sltrm5* mutant (Fig. 4c–e; Supplementary Table 5). To reveal the mechanisms that underlie the histology of shape, cellular parameters at the proximal end of the anthesis-stage ovaries were characterized. In the *ovate* and *ovate/sov1* genotypes, cell number increased significantly along the proximo-distal axis and decreased significantly along the medio-lateral axis. These cell number changes were offset in the presence of *Sltrm5*, resulting in cell numbers that approximated the values found in wild-type ovaries, especially in the proximo-distal direction (Fig. 4e; Supplementary Table 6). Cell length and width also increased in *ovate* and *ovate/sov1*, and *Sltrm5* rescued these traits in these mutant backgrounds. However, cell shape indices were highly variable and barely altered compared to wild-type, suggesting a minor role for cell elongation in regulating ovary length. Together, these data show that *OVATE* and *SlOFP20* synergistically function in shape regulation by primarily modulating cell division patterns. The critical cell division patterns occur during floral development and perhaps as early as when carpel primordia arise. *SlTRM5* balances the *OVATE* and *SlOFP20* division patterns to determine final tomato shapes. Combined with the protein localization patterns

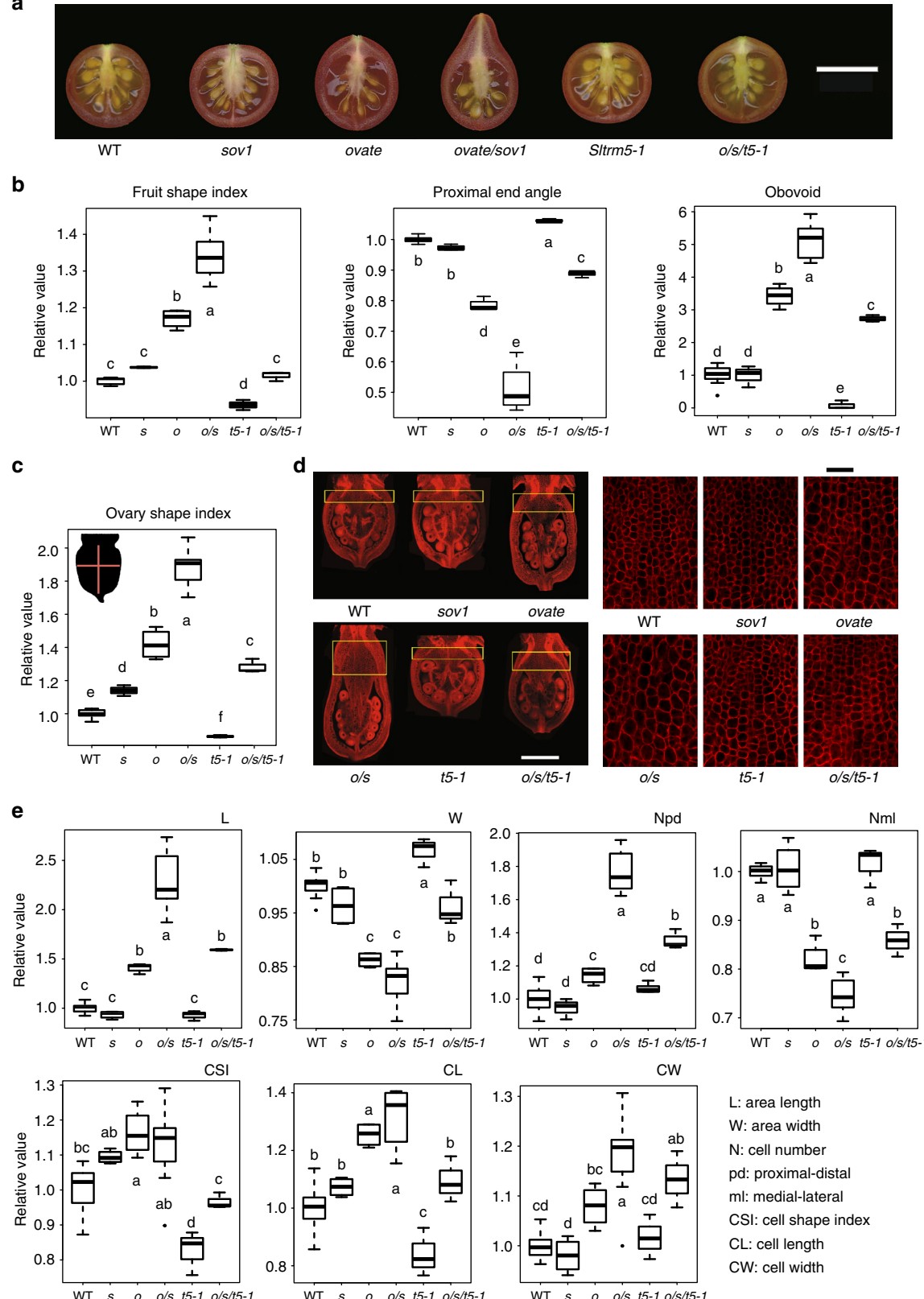

in tobacco, our data suggest that the relocalization of the OFP-TRM protein complexes are critical to fine tune cell division patterns early in development, thereby ultimately controlling fruit shape. However, these protein relocalizations would have to be further investigated in growing organ primordia of the plant.

**A common pathway regulating plant organ shapes.** Whereas the effects of natural allelic variation in tomato TRMs are unknown, *ovate* and *sov1* have a strong impact on fruit shape in the NILs (Fig. 4a). To determine when the *ovate* and *sov1* alleles arose, we evaluated their presence in ancestral populations that

**Fig. 4** Complementation of fruit and ovary elongation by *Sltrm5* in the *ovate/sov1* background. **a** Mature fruits of the *ovate/sov1/Sltrm5* single, double and triple NILs. *o*, *ovate*; *s*, *sov1*; *t5-1*, *Sltrm5-1*. **b** Comparisons of mature fruit traits among the *ovate/sov1/Sltrm5* NILs. Fruit shape index is the ratio of length to width; proximal end angle is the angle of the fruit at 10% from the very top of the fruit; obovoid is the area below the mid-point of the fruit. **c** Ovary shape index at anthesis in the different NILs. **d** Propidium Iodine staining of anthesis-stage ovaries (left) and cell sizes in the proximal end of anthesis ovaries (right). The proximal area of an ovary is indicated by the yellow outline of the box and refers to the length and width traits measured in **e**. **e** Histological and cellular evaluations of the proximal end of anthesis ovaries in the different NILs. The letters in the boxplots indicate the significant differences among different genotypes evaluated by Duncan's test ($\alpha < 0.05$). The lower and upper bounds of the box in a boxplot indicate the first and third quartiles, respectively. The median and outliers are indicated by the center line and dots, respectively. The scale bars for fruit, ovary, and cell morphology represent 1 cm, 0.5 mm, and 50 μm, respectively

were collected in Peru and Ecuador (first domestication site), and Mexico (second domestication site)[16]. The data showed that *ovate* and *sov1* arose in the ancestral semi-domesticated germplasm of tomato, and that both mutant alleles were found in low frequency, yet never in the same accession that resulted from these early domestication events (Fig. 5a). We determined whether both mutations are required for obovoid shape in modern day tomatoes. The data showed that the mutant alleles at these loci combined after initial domestication since every *ovate/sov1* accession produced obovoid fruits (Fig. 5b, c; Supplementary Data 3). Obovoid accessions that only carried the mutation at *ovate* were equally elongated in shape but showed a reduced degree of obovoid, and an increase in proximal end angle as well as eccentricity (Fig. 5d). Combined, these morphological attributes demonstrate that most extreme obovoid tomato fruit requires mutations in both *OVATE* and *SlOFP20*.

Our results clearly demonstrate the critical role of two OFP proteins in regulating tomato fruit shape by changing cell division patterns early in the development of the organ. The role of OFPs in regulating the shapes of other plant organs and in other species is less clear. Melon (*Cucumis melo*) fruits are morphologically diverse and several QTLs associated with shape have been identified[17,18]. We fine-mapped the fruit shape QTL *fsqs8.1* in a population derived from a cross between Piel de Sapo and PI124112, to a 152 kb segment on melon chromosome 8 carrying just six candidate genes including *CmOFP13* (Fig. 6a; Supplementary Table 7). In cucumber (*Cucumis sativus*), a species closely related to melon, the fruit shape QTL *fs3.2* interval contains *CsOFP15*, a gene that is lower expressed in cucumber varieties with an elongated fruit[19]. This is expected if *CsOFP15* controls cucumber fruit shape. Tuber shape in potato (*Solanum tuberosum* L.) and its wild relatives (*Solanum* section *Petota*) varies from very round to long and narrow such as in fingerling varieties. The tuber shape QTL *Ro* has been fine-mapped in an outcrossing $F_1$ population to a region on potato chromosome 10 that appears syntenic to the tomato chromosome region encompassing *SlOFP20*[20,21] (Fig. 6b; Supplementary Table 8). However, the ortholog of tomato *OFP20* is not present in the genome of potato DM1–3 at this location. On the other hand, a round tuber parent, M6, carries an ortholog of *SlOFP20* (Fig. 6b). Using a diploid potato $F_2$ population derived from a cross between the round tuber parent, M6 and the parent producing elongated tubers, DM1–3, showed a strong association with the *StOFP20* marker suggesting that the gene is controlling tuber morphology in this population (Fig. 6b; Supplementary Fig. 12a)[22]. Mapping the DM1–3 genomic reads to the M6 genome showed a deletion of approximately 30 kb in DM1–3 including *StOFP20* (Supplementary Fig. 12b), explaining the absence of this gene in the DM1–3 reference genome of potato. These data support the notion that potato tuber shape mapped to the *Ro* locus is controlled by the ortholog of tomato *OFP20*. OFPs also impact the shape of other organs. Overexpression of *AtOFP1* in Arabidopsis leads to shorter leaves and floral organs[5], which is similar to overexpression of *SlOFP20* in tomato leading to

rounder leaflets and shorter leaves (Fig. 1c). Thus, OFPs regulate shapes of many if not all organs in plants.

In tomato, we showed that the null mutation in *SlTRM5* genetically interacted with *ovate* and *sov1* to suppress the phenotype to a round shape primarily by reducing cell division in the proximo-distal direction. In cucumber, a population derived from a cross between WI7238 and WI7239 carrying long and round fruit, respectively, led to the identification of two fruit shape QTL *fs1.2*, and *fs2.1*[23]. We fine-mapped the cucumber fruit shape QTL, *fs2.1* to 10 candidate genes including an ortholog of *AtTRM5/SlTRM5* (Fig. 6c; Supplementary Table 9). Mutations in *GL7/GW7* encoding a rice TRM that belongs to the Arabidopsis TRM1–5 (*LONGIFOLIA*) clade (Supplementary Fig. 13a) affects grain shape by changing cell division and cell elongation patterns[8–10]. Additionally, overexpression or loss-of-function mutants of *AtTRM1* and *AtTRM2* in Arabidopsis produce elongated or shortened siliques and leaves, respectively, resulting from altered cell elongation[7,11]. In addition to fruit shape, *Sltrm5* also features rounder leaflets (Supplementary Fig. 11b), producing shapes that are similar as found in *SlOFP20* overexpressors (Fig. 1c). The results suggest that collectively certain TRMs control cell number and/or cell shape, and influence organ shapes in a similar manner as certain OFPs. Intriguingly, most of the genetically identified *OFPs* and *TRMs* that control plant organ shape fall in the Arabidopsis *OFP1–5* and *TRM1–5* clades respectively (Supplementary Fig. 13). This finding implies that orthologous members of these subclades are likely involved in regulating organ morphology in many other plant species.

## Discussion
In addition to providing evidence that *OFPs* and *TRMs* control fruit, tuber, vegetable, and grain shapes in domesticated plants and were the targets of selection, our findings uncovered a link between these two families which are common in genomes of multicellular plants[11,24]. This finding extends to other plant parts such as leaves and floral organs in model and crop plants, and span both eudicot and monocot species. A model for genetic mechanism of OFPs and TRMs in organ formation implies that their relative expression levels are critical to control the eventual shape of plant organs (Fig. 7). We propose that the balance between subcellular localization of the protein complexes, e.g., in the cytoplasm or associated with microtubules, may determine the growth patterns underlying organ shape. Moreover, the relocalization of OFPs and TRMs to different subcellular compartments upon interaction suggests that a dynamic balance between cytoplasmic-localized and microtubular-localized OFP-TRM protein complexes may regulate cell division and growth early in development of the organs. These results offer opportunities for plant breeders to target selection and/or genome editing approaches to create varieties for niche markets as well as for crop improvement in general. In addition, the apparent universality of the OFP-TRM module is likely to be part of a network required for coordinated multicellular growth in all plants.

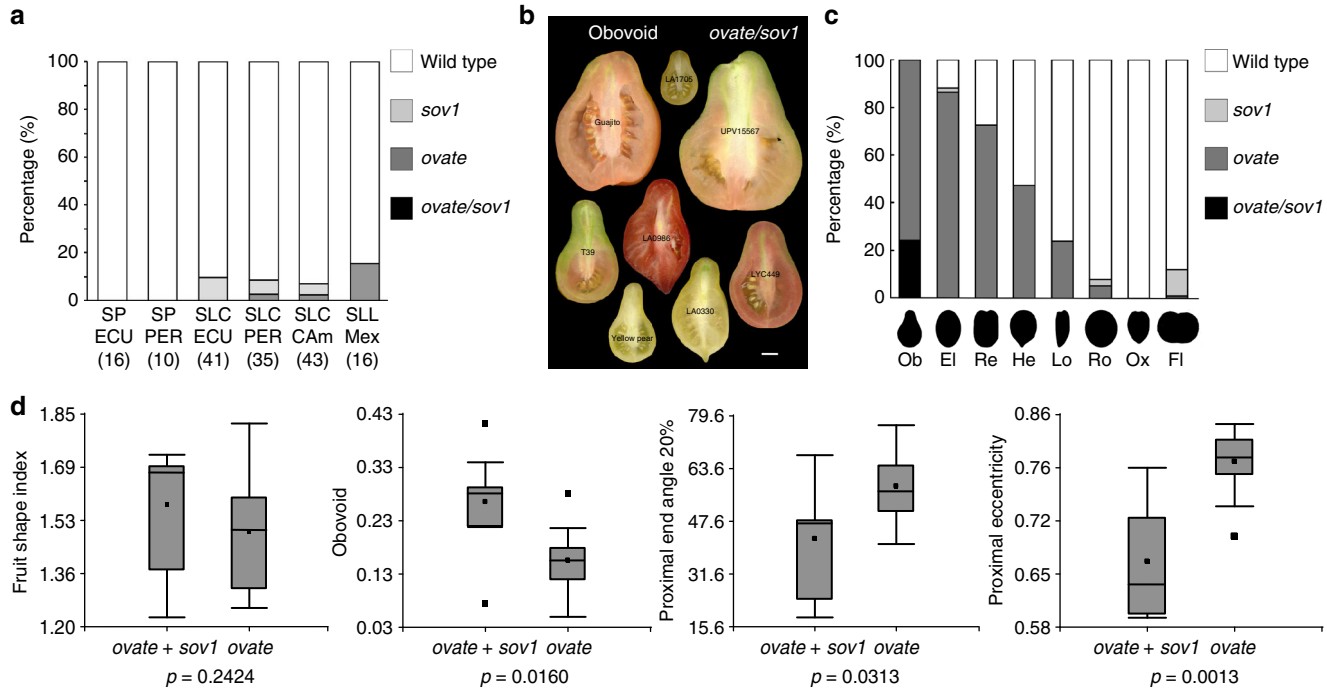

**Fig. 5** Distribution of *sov1* and *ovate* in tomato germplasm and its impact on fruit morphology. **a** Frequency of *ovate* and *sov1* in an ancestral population of tomato. SP ECU, *Solanum pimpinellifolium* accessions from Ecuador; SP PER, *Solanum pimpinellifolium* accessions from Peru; SLC ECU, *Solanum lycopersicum* var. *cerasiforme* accessions from Ecuador; SLC PER, *Solanum lycopersicum* var. *cerasiforme* accessions from Peru, SLC CAm, *Solanum lycopersicum* var. *cerasiforme* accessions from Central America; SLL MEX, *Solanum lycopersicum* var. *lycopersicum* accessions from Mexico. The number of accession in each category is indicated in parentheses. **b** Shape of fruits from cultivated tomato accessions carrying both the *ovate* and *sov1* mutations. The scale bar represents 1 cm. **c** Frequencies of *ovate* and *sov1* in different tomato shape categories in percentage based on the number of accessions in the category. Shape categories are: obovoid (Ob), ellipsoid (El), rectangular (Re), heart (He), long (Lo), round (Ro), oxheart (Ox) and flat (Fl), respectively. **d** Variation among fruit shape traits of obovoid fruits for cultivars carrying *ovate* or *ovate/sov1* mutations. The *p* value represents the result of the student's *t*-test between groups

## Methods

**DNA sequencing and marker development**. DNA was extracted from Yellow Pear (YP) and Gold Ball Livingston (GBL) leaf samples using the Qiagen kit (Germantown, MD) and following manufacturer's instructions. DNA libraries were constructed and sequenced on the Illumina HiSeq2000 platform with the 101-bp paired-end mode at the Genome Technology Access Center (GTAC) at Washington University, St Louis MO (SRA SRP127270). After removing adapters and low-quality sequences, the sequences represented more than 20× coverage of the genome (aligned bases divided by 950 Mbp length of the tomato genome). High quality cleaned reads were aligned to the tomato genome SL2.40[25] using NovoA-lign (http://www.novocraft.com/). SNPs were identified in the genomic regions of interest using SAMtools[26]. High quality (>100) homozygous SNPs with coverage higher than 15× were kept for marker development. Based on the identified SNPs, we developed 26 additional dCAPS markers located between the existing markers, 12EP153 and 12EP5 (Supplementary Data 4) using dCAPS Finder (http://helix.wustl.edu/dcaps/dcaps.html) and Primer 3 (http://bioinfo.ut.ee/primer3-0.4.0/). The *sov1* deletion in the promoter of *SlOFP20* was identified by aligning the reads of the YP and GBL to the reference genome (Supplementary Fig. 2). After validation by Southern blotting, the structural rearrangement was further validated using primers 13EP549, 13EP550 and 13EP551 in a single PCR reaction. The wild-type band runs at 1220 bp whereas the fragment that amplifies YP DNA runs at 900 bp. The Heinz 1706 BAC clone sequence, LE_HBa0040P16 containing *SlOFP20* (Genbank AC244530) allowed us to estimate the size of the *sov1* deletion to 31 kb and its position in relation to the coding region of *SlOFP20*. The estimation of putative transcription start site of *SlOFP20* is based on the tomato ESTs and cDNAs database (https://solgenomics.net/) as well as our RNA-seq data (SRA SRP090032; SRP089970).

**Potato *StOFP20*-trait association**. The M6 potato genome scaffold 814 harbored the ortholog of *SlOFP20*[27]. Primers were designed to amplify the ortholog in M6 and a linked sequence in DM1–3 (Supplementary Data 4). The marker was genotyped using 190 individuals of a segregating F$_2$ population derived from DM1–3 and M6[22]. ANOVA analyses were applied to determine the significance of the association of the marker allele with the phenotype.

**Alignment of the potato DM1–3 reads to the M6 scaffold 814**. Reads from the 16 paired-end genomic DNA libraries of potato DM1–3 used in the de novo assembly of the reference genome (SRA029323) were aligned to M6 scaffold 814 using BWA-MEM v0.7.13[28] in paired-end mode with default parameters. Alignments with MAPQ score higher than or equal to 30 were kept and visualized with Integrative Genomics Viewer (IGV v2.3.68).

**Overexpression of *SlOFP20***. Full length *SlOFP20* was amplified from DNA using primers 13EP617/618 (Supplementary Data 4) and Phusion® high-fidelity DNA polymerase (New England Biolabs, Ipswich, MA). The fragment was cloned into the *XhoI/SacI* site of the pKYLX71 vector[29] and expressed under the 35S promoter. The construct was sequenced to confirm that no errors were introduced during the amplification and then transformed in YP. Five independent transgenic lines were obtained and fruit and leaf shape analysis was done using 8 mature fruits or primary leaflets of each plant.

**Downregulation of *SlOFP20***. Two artificial microRNAs (amiRNAs) targeting the coding region of *SlOFP20* were designed using the WMD3 Web MicroRNA Designer (http://wmd3.weigelworld.org/) and put into the Arabidopsis *MIR319a* precursor backbone. The sequences of amiRNAs designed to knock-down the expression of *SlOFP20* were provided in Supplementary Data 4. The engineered amiRNA precursors were synthesized at GenScript Biotech Corporation (Piscataway, NJ) and then cloned into the *SacI/XbaI* site of the pKYLX71 vector[29]. The two *SlOFP20* amiRNA constructs were transformed in LA1589 resulting in 12 primary transgenic lines. The T$_0$ lines were evaluated for downregulation of *SlOFP20* using semi-quantitative PCR. Two of the most down regulated lines per construct were backcrossed to SA29 carrying the *ovate* mutation. The resulting seedlings were selected for the presence of the transgene using primers EP553/554 (Supplementary Data 4) and selfed to produce families 15S91, 15S92, 15S93, and 15S94 (BC$_1$F$_2$).

***SlTRM* CRISPR/Cas9 mutants**. A CRISPR/Cas9 construct was designed to create mutations in both *SlTRM3/4* and/or *SlTRM5*. The construct was assembled using the Golden Gate cloning method[30]. Two sgRNAs specifically targeting each of the

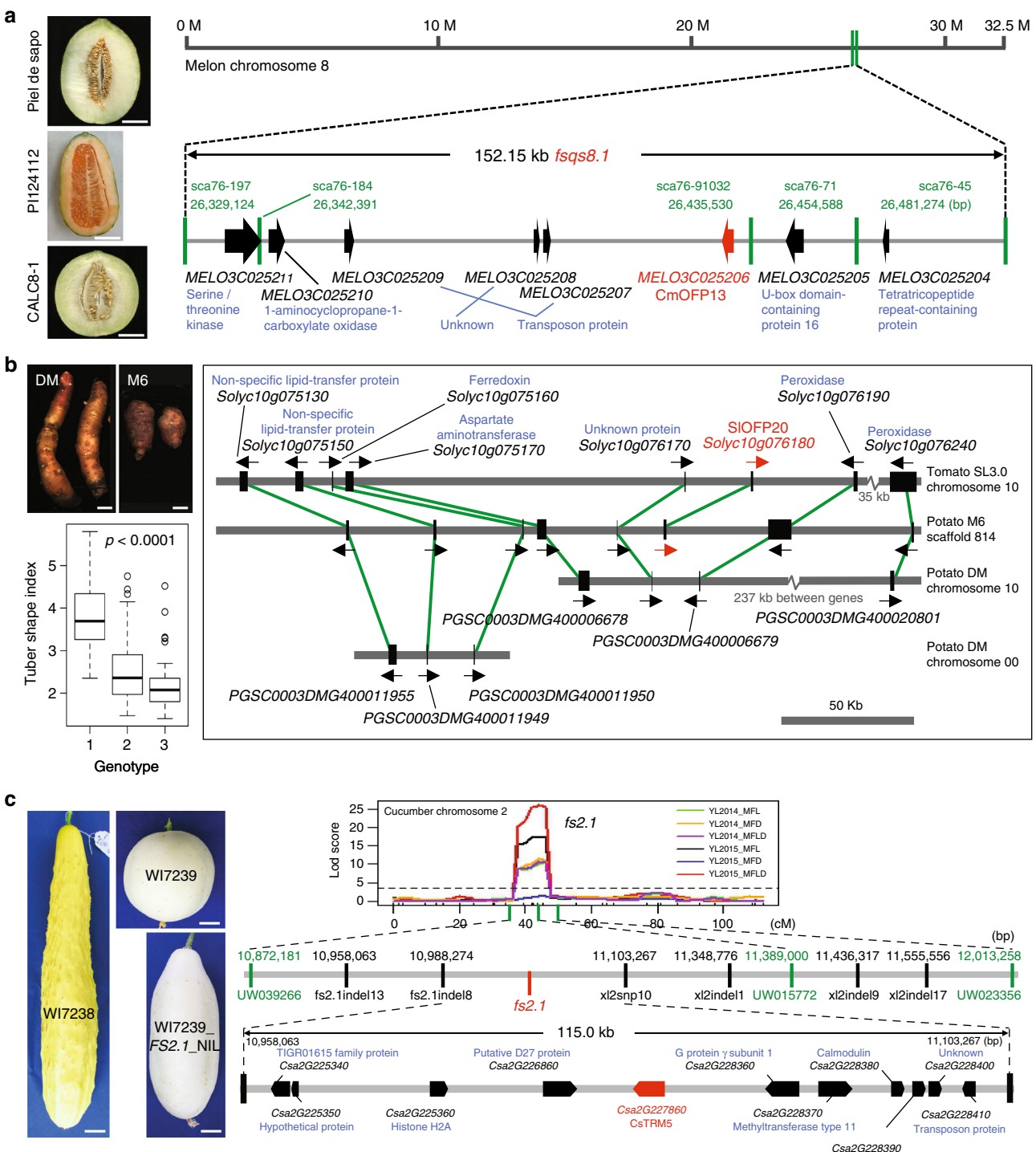

**Fig. 6** Association of OFP and TRM in regulating fruit and tuber shape in domesticated crops. **a** Fine-mapping of *fsqs8.1* in melon to *CmOFP13*. Fruit shapes of the parental lines and the CALC8-1 NIL that carries the PI124112 allele of *fsqs8.1* in the Piel de Sapo background. Scale bar, 5 cm. **b** Association of tuber shape with *StOFP20* in potato. 1, homozygous DM1-3 allele; 2, heterozygous; 3, homozygous M6 allele. The green lines connect the syntenic orthologous gene pairs in tomato and the two potato accessions around *SlOFP20*. Scale bar, 1 cm. **c** Fine-mapping of the cucumber *fs2.1* locus. The LOD curve from the cucumber F$_2$ population led to candidate gene identification including *CsTRM5*, an ortholog of *SlTRM5*. Likely candidate genes are denoted in red. Other gene annotations are denoted in blue. Scale bar, 2.5 cm

two *SlTRM*s were amplified using the pICH86966::AtU6p::sgRNA_PDS construct (Addgene plasmid #46966, www.addgene.org) as a template with the reverse primer: 13EP639 and forward primers: 14EP292 and 14EP294 for *SlTRM3/4* and *SlTRM5*, respectively (Supplementary Data 4). The level 1 constructs pICH47751 (Addgene #48002) and pICH47761 (Addgene #48003) were assembled using the level 0 construct pICSL01009::AtU6p (Addgene #46968) and the sgRNA PCR

products to place each sgRNA under the Arabidopsis U6 promotor. Level 1 constructs, pICH47732::NOSp::NPTII (Addgene #51144), pICH47742::35S::Cas9 (Addgene #49771), pICH47751::AtU6p::sgRNA-*SlTRM3/4*, pICH47761::AtU6p:: sgRNA-*SlTRM5* and the linker pICH41780 (Addgene #48019) were then assembled into the level 2 vector pAGM4723 (Addgene #48015). All the vectors for building the CRISPR-Cas9 construct were provided by Vladimir Nekrasov

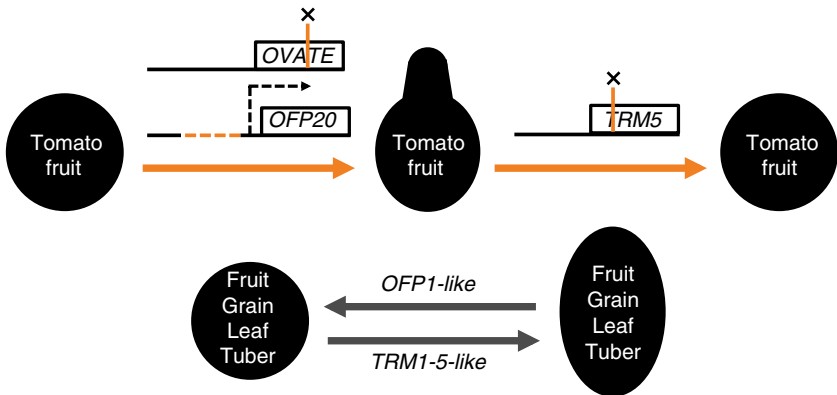

**Fig. 7** The OFP-TRM module that regulates plant organ shape. The natural derived allele of *OVATE*, a true null, and the natural derived allele of *SlOFP20*, a deletion in the upstream regulatory region proposed to reduce expression of the gene, result in pear-shape tomato fruit. A CRISPR-Cas9 induced frame shift mutation in *SlTRM5* rescues the fruit shape phenotype to a nearly round fruit. In general, the TRM1–5-like genes promote the elongation of fruit, grain, leaf and tuber, whereas OFP1-like genes play an antagonistic role

and Sophien Kamoun, The Sainsbury Laboratory, Norwich Research Park, Norwich, UK.

**Plant transformation and genotyping**. Constructs were transformed into tomato at the Plant Transformation Facility at University of California (Davis, CA 95616) (amiRNA and overexpressor constructs), and by Dr. Joyce Van Eck, Cornell University (CRISPR-Cas9 constructs). The lines overexpressing or under-expressing *SlOFP20* were genotyped for the presence of the construct with primers designed to amplify the *NPTII* gene conferring kanamycin resistance (EP553/554; Supplementary Data 4). For the CRISPR-Cas9 constructs, we received 16 independent $T_0$ lines. By sequence analyses, the majority were found to carry mutations in both *SlTRM3/4* and *SlTRM5* and occasionally mutations were in both alleles of one or both genes. Most lines were sterile (male and female) as no fruits with seeds were generated by selfing or with wild type pollen onto mutant styles or mutant pollen unto wild type styles. The primary transformants that were fertile (four $T_0$ lines) carried an in frame deletion (multiples of 3) and wild type allele for *SlTRM3/4* and an in frame or frameshift mutations for *SlTRM5*. These fertile lines were crossed to LA1589 in order to remove the *Cas9* transgene and stabilize the lines. Once stabilized, for *SlTRM5* the 1 bp deletion and 1 bp insertion alleles (causing a frameshift mutation and likely a null) were backcrossed into the *sov1/ovate* background, which is described above for the development of the NILs.

**Fruit shape analysis**. Full size maturing fruits were cut longitudinally, scanned at 300 dpi, and analyzed using Tomato Analyzer v3.0[31,32]. The following attributes were measured in most samples: fruit shape index, proximal end angle, proximal eccentricity and obovoid, which measures the pear-shapedness. Fruit Shape Index is the ratio of the maximum height length to maximum width of a fruit. Proximal end angle is the angle between best-fit lines drawn through the fruit perimeter on either side of the proximal end point at 10% (NILs) or 20% (tomato varieties). Proximal eccentricity is the ratio of the height of the internal ellipse (defined by the section of the fruit where the seeds are located) to the distance between the bottom of the ellipse and the top of the fruit. Obovoid is calculated as the maximum width (W), the height at which the maximum width occurs (y), the average width above that height (w1), and the average width below that height (w2) are calculated and a scaling function scale_ob is used for calculation: Obovoid = 1/2 × scale_ob (y) × (1 − w1/W + w2/W), if obovoid > 0, subtract 0.4; otherwise, obovoid is 0. For the potato tuber shape analyses, length and width were measured in Image J and shape index was calculated by taking the ratio of these two measurements. The length of the curved potatoes was measured by tracing the curve. Melon Fruit Shape Index was calculated with Tomato Analyzer 3.0 from scanned images. For the cucumber shape analyses, length and width were measured with calipers and the shape index was calculated by taking the ratio of the two.

**Ovary shape analysis**. Anthesis ovaries were cut longitudinally and digitalized using the Olympus Szx9 (SZX-ILLB2–100) dissecting scope. The maximum length and width of ovaries were measured using ImageJ software (https://imagej.nih.gov/ij/), and from this, the ratios of the maximum length to maximum width were calculated. Three to five plants of each genotype were analyzed, and the average values each taken from 8 to 10 fruits or ovaries per plant were analyzed with Tukey's HSD test ($\alpha < 0.05$).

**Cellular attributes**. Anthesis ovaries were cut longitudinally with a razor blade and fixed in FAA (50% Ethanol, 10% Formaldehyde, 5% glacial acetic acid) at 4 °C

overnight. The samples were dehydrated on ice with ethanol-ddH$_2$O series (50, 70, 85, 95, 100% x 2), and rehydrated with ethanol-ddH$_2$O series (95, 85, 70, 50, 30, 15%) at one hour for each step. The samples were rinsed with ddH$_2$O twice for 20 min each. Ovaries were incubated on ice in the staining buffer (0.02 mg/ml propidium iodide, (MP Biomedicals), 0.02% DMSO) for 1 h and rinsed with ddH$_2$O for 20 min twice, then dehydrated with ethanol-ddH$_2$O series (50, 70, 85, 95, 100%) on ice, and 100% ethanol at room temperature overnight. Finally, the samples were treated with 1:1 ethanol: methyl salicylate for 1–2 h followed by 100% methyl salicylate (Fisher Chemical) at 4 °C for 2–3 days. The sections were imaged using a Zeiss LSM 510 META Confocal Microscope, and cell size and number assessments were made using the ImageJ software package. The proximal area of the ovary (the area above the ovules closest to the stem end of the flower) was used for the histological and cellular analysis. The length and width of the entire proximal area was measured. Numbers of parenchyma cells were counted in the middle of the proximal area in both the proximo-distal and medio-lateral direction. The length and width of parenchyma cells in the proximal area were evaluated on at least 20 cells.

**Yeast two-hybrid analyses**. Full-length *OVATE* or full length *SlOFP20* were cloned into the pGBKT7 vector (Clontech, Mountain View, CA) as C-terminal fusions to the GAL4 DNA binding domain (BD). Fragments of *SlTRM5*, *SlTRM17/20a* and *SlTRM25* were cloned into the pP6 vector (Hybrigenics, Paris, France) as a C-terminal fusion to the GAL4 activation domain (AD). A pair of bait and prey plasmids were transformed into the Y2HGold yeast strain following the Clontech Yeast Protocol Handbook instructions (Clontech, Mountain View, CA) and plated on minimal medium lacking Trp and Leu (SD/-Trp/-Leu) with X-α-Gal (the substrate of *MEL-1* gene product, α-galactosidase). Quantification of reporter α-galactosidase activity was performed using the Clontech Yeast α-Galactosidase Assay Kit. Three single colonies of each combination were grown in liquid synthetic dropout medium lacking leucine and tryptophan at 30 °C with shaking (250 cycles/min) for 18 h. After recording the OD$_{600}$, the supernatants were collected via centrifugation at 18,800$xg$ for 2 min. After adding the assay buffer (PNP-α-Gal + CH$_3$COONa) to the supernatants and incubating at 30 °C for 60 min, the reaction was terminated by adding the stop solution (Na$_2$CO$_3$). The optical density of each sample was recorded at OD$_{410}$ and α-galactosidase activity was calculated using the formula: $1{,}000 \times Vf \times OD_{410}/[(e \times b) \times t \times Vi \times OD_{600}]$ where $t =$ time (min) of incubation, $Vf =$ volume of assay (200 or 992 µl), $Vi =$ volume of culture medium supernatant added, OD$_{410} =$ A410 of the reaction mix, OD$_{600} =$ A600 of 1 ml of culture, $e \times b = p$-nitrophenol molar absorbtivity at 410 nm × the light path (10.5 (ml/µmol) for 200-µl format = 16.9 (ml/µmol) for 1-ml format where $b = 1$ cm). The empty vector was used as the negative control.

**Phylogenetic and protein motif analyses of TRMs**. To retrieve all putative SlTRM proteins in tomato, full-length sequences of 34 Arabidopsis members of this family were used for BLAST similarity search against the International Tomato Annotation Group release 2.3 predicted proteins (ITAG 2.40) (http://solgenomics.net/) and Motif Alignment and Search Tool (MAST) search[33], both at a cutoff E-value of $10^{-5}$. The MEME tool[34] was used to define the conserved motifs with the following parameters: "nmotifs 8, minw 10, maxw 100, minsites 30, maxsites 120". ClustalW was used for multiple sequence alignment procedures. The phylogenetic relationships among the all the TRMs in tomato and Arabidopsis were estimated with neighbor-joining method based on the p-distance (i.e., the phylogenetic distances were obtained by dividing the number of amino acid differences with the total number of sites compared) and 1000 bootstrap validation. The phylogenetic

tree was visualized by FigTree (http://tree.bio.ed.ac.uk/software/figtree/). Protein charge plot were generated by calculating the total charge of amino acids over a sliding window of 51 residues as described previously[11].

**Phylogenetic analysis of the *SlOFP20* subclade**. We retrieved the potato ortholog and closest paralogs (four total) by conducting reciprocal best BLAST hits. This led to four OFP members in the M6[35] and doubled monoploid Group Phureja clone DM1–3 516 R44 (DM)[36,37]. The four hits in each potato assembly with the lowest E-value statistic were identified as initial candidate sequences. In order to find the best full-length tomato protein BLAST hits of each potato candidate sequence, the amino acid sequence returned for each hit was used as query in a BLASTp search against the SL3.0 tomato annotated protein sequences. Reciprocal best BLAST hit analysis was performed using these full-length tomato proteins. Each full-length protein was used as query in tBLASTn searches against the DM1–3 and M6 assemblies. The best DM1–3 and M6 hits from each of those searches was then used as query in reciprocal BLASTp and tBLASTn searches against the tomato protein sequences and genome SL3.0 assembly respectively, in order to identify and confirm each reciprocal best BLAST hit relationship. BLAST searches used the BLOSUM62 scoring matrix and a word size of 11 amino acids. Using similar methods, we obtained the five Arabidopsis OFP1 through OFP5 and the four melon OFP proteins that showed the best BLAST hit relationship to members of the SlOFP20 subclade. We also included OVATE and its best BLAST hit AtOFP7. The resulting reciprocal best BLAST hits were aligned with ClustalW. Phylogenetic relationships between them were estimated via the neighbor-joining method and p-distance, with the phylogeny rooted by the distant Physcomitrella OFP protein and validated with 1000 bootstraps, as implemented in Geneious 10.1.3[38].

**Phylogenetic analysis of the *AtTRM1–5* clade**. We used the five Arabidopsis and three tomato proteins from Supplementary Fig. 4. We identified the rice and cucumber *TRMs* as described for the *OFP1–5* clade. Phylogenetic analysis was implemented similarly as with the *OFP* reciprocal best BLAST hits, but with the phylogeny rooted by the Solyc02g089050 TRM protein.

**In vitro mutagenesis**. Site-directed mutagenesis of selected residues for *SlTRMs* and *OVATE/SlOFP20* was carried out with the QuikChange II XL Site-Directed Mutagenesis system (Agilent Technologies, Santa Clara, CA) according to the manufacturer's specifications. Oligonucleotide primers between 25 and 35 bases with a melting temperature of ≥78 °C were designed using quikchange primer design software (http://www.genomics.agilent.com/primerDesignProgram.jsp) to cover the appropriate point mutations that would lead to the desired amino acids substitutions (Supplementary Data 4). The *SlTRMs* and *OVATE/SlOFP20* mutants were subcloned into pP6 and pGBKT7 vectors respectively (Clontech, Mountain View, CA). Mutations were verified by DNA sequence analysis of the resulting clones.

**Construction of plasmids for transient transformation**. For constructs used in the transient assays, full-length wild type or mutant coding sequences (CDS) of *OVATE*, *SlOFP20*, *SlTRM3/4*, *SlTRM5* and *SlTRM25* were cloned into *pENTR/D-TOPO* Gateway entry vector (Invitrogen, Carlsbad, CA) following the manufacturer's protocol. The coding regions were recombined into binary destination expression vectors pH7RWG2 (Cauliflower mosaic virus (CaMV) 35S promoter-driven) for C-terminal RFP (red fluorescent protein) fusions[39,40] or pSITE-2CA and pSITE-2NA (2 × 35S) for N-terminal and C-terminal GFP (green fluorescent protein) fusions, respectively[41]. For the BiFluorescence Complementation (BiFC) experiments, full length wild-type *OVATE* and *SlTRM25* were cloned as C-terminal or N-terminal fusions to the C-terminal or N-terminal fragment of the Yellow Fluorescent Protein (YFP) in the BiFC vectors pYN-1 (N-terminal fusion of YFP$_{1-158}$), pYC-1 (N-terminal fusion of YFP$_{159-238}$), p2YN (C-terminal fusion of YFP$_{1-158}$) and p2YC (C-terminal fusion of YFP$_{159-238}$)[42] and co-expressed in all 8 combinations in *N. benthamiana* to determine the best usage of the BiFC vectors. Based on the overall YFP intensity and numbers of cells expressing YFP, the pYN-1 and p2YC vectors were used for *SlTRMs* and *OVATE*, respectively.

**Transient expression of proteins in *N. benthamiana***. *Agrobacterium tumefaciens* strain C58C1 was used for the transient transformations. Colonies carrying the binary plasmids were grown at 28 °C on LB medium plates that contained 50 µg/ml gentamycin and 25 µg/ml rifampicin for selection of the strain, and 100 µg/ml spectinomycin for selection of the binary vectors, pH7RWG2 (carrying *OVATE* or *SlOFP20* with a C-terminal RFP tag), pSITE-2CA (carrying *SlTRM3/4* or *SlTRM25* with a N-terminal GFP tag), and pSITE-2NA (carrying *SlOFP20* with a C-terminal GFP tag)[39–41]. For agroinfiltration, single colonies were grown in liquid LB supplemented with gentamycin, rifampicin and spectinomycin overnight (28 °C, 220 cycles per minute). Fifty µl of the agro suspension was added to 5 ml LB with the same antibiotics for another overnight incubation under the same conditions. The agrobacteria were pelleted by centrifuging at 1610×*g* for 20 min or 2200×*g* for 15 min. The cells were resuspended in infiltration buffer containing 10 mM MgCl$_2$, 10 mM MES, PH 5.7 and 150 mM acetosyringone at pH 5.6 and adjusted to an OD$_{600}$ of 0.2–0.3. The cells were incubated at room temperature for 3 h without shaking prior to infiltration. To enhance transient expression of the fusion proteins, the viral suppressor of gene silencing p19 protein was coexpressed in most of the experiments. For co-infiltration, equal volumes of cultures were mixed and infiltrated into *N. benthamiana* leaves through the abaxial surface using a 1-ml needleless syringe (Becton, Dickinson and Company). Plants were then kept in a growth room at 24/22 °C with a 16/8 h light/dark photoperiod for 48–72 h.

**Epifluorescence and confocal microscopy**. *N. benthamiana* leaf samples (approximately 0.25 cm$^2$ near the infiltrated area) were collected at 70–90 h post-infiltration, mounted in water and viewed directly with a Zeiss LSM 880 confocal scanning microscope using an oil immersion objective 40× Plan-Apochromat 1.4NA (numerical aperture of 1.4). Fluorescence was excited using 488 nm and 543 nm light for GFP and RFP, respectively. GFP and RFP emission fluorescence was selectively detected at 490–540 and 550–630 nm using the Zen 2.3 SP1 software. For each experiment, 50–100 cells in two independent leaves that expressed both proteins were evaluated. For the BiFC experiments, YFP fluorescence was excited using 514 nm laser and detected at 520–550 nm using the Leica TCS SP5 confocal scanning microscope with a 25x objective.

**Reporting Summary**. Further information on research design is available in the Nature Research Reporting Summary linked to this article.

## Data availability

Raw sequence reads have been deposited into the DNA DataBank of Japan (DDBJ) under accessions SRP090032; SRP089970; SRA061767 for the mRNA seq data; and SRP127270 for the whole genome sequence data. The normalized expression data are also available at the Tomato Functional Genomics Database (TFGD; http://ted.bti.cornell.edu/) under the experiments D015 and D016. A Reporting Summary for this Article is available as a Supplementary Information file. Other methods can be found in the Supplementary Methods section in the Supplementary Information file.

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

## Acknowledgements

We thank Drs. David Bouchez and Martine Pastuglia (INRA, Versailles, France) for helpful discussions and suggestions. Jiheun Cho for greenhouse care and Jason Van Houten (Ohio State University) for technical support. Dr. Zhangjun Fei (USDA-ARS Ithaca NY) for helpful comments and suggestions. Drs. Wolfgang Lukowitz, Bob Schmitz and Kelly Dawe (University of Georgia) and Dr. Cris Kuhlemeier (University of Bern, Switzerland) for helpful comments on the manuscript. Funding for this research has been provided by National Science Foundation IOS 0922661, the Ohio State Tornado Recovery Fund OHOA0499, Ohio Agricultural Research and Development Center SEEDS grant 2014018, University of Georgia 1021RX070014, USDA AFRI National Institute of Food and Agriculture 2017-67013-26199 (EvdK). Ohio Agricultural Research and Development Center SEEDS grant 2014084 (SW). The Spanish Ministry of Economy and Competitiveness/FEDER grant AGL2015-64625-C2-2-R (AJM). The USDA AFRI National Institute of Food and Agriculture 2015-51181-24285 (YW). The USDA AFRI National Institute of Food and Agriculture 2014-67013-22434 (SHJ).

## Author contributions

S.W.: co-wrote the original draft, conceptualization, formal analysis, investigation, methodology, validation and visualization. B.Y.: review and editing, formal analysis, methodology, investigation, validation. N.K.: review and editing, formal analysis, methodology, investigation, visualization, validation. G.R.R.: review and editing, formal analysis, investigation, visualization. H.J.K.: review and editing, formal analysis. M.C.: review and editing, methodology. EIB: review and editing, formal analysis, data curation. N.K.T.: review and editing, investigation, methodology, visualization. M.J.G.: review and editing, formal analysis. A.D.: review and editing, formal analysis. Y.P.: review and editing, formal analysis, investigation. C.L.: review and editing, sharing sequence data prior to publication. C.R.B.: review and editing, sharing sequence data prior to publication. D.H.: review and editing, formal analysis. Y.W.: review and editing, formal analysis, funding acquisition, investigation. S.H.J.: review and editing, formal analysis, funding acquisition, investigation. H.V.E.: review and editing, formal analysis, investigation, methodology. J. W.: review and editing, formal analysis, investigation, methodology. A.J.M.: review and editing, formal analysis, funding acquisition, investigation. T.M.: review and editing, formal analysis, methodology, investigation. E.vdK.: co-wrote the original draft, formal analysis, funding acquisition, conceptualization, investigation, project administration, supervision.

## Additional information

**Competing interests:** The authors declare no competing interests.

