## [Peer Review File · Nature Communications]

Reviewers' comments:

Reviewer #1 (Remarks to the Author):

Manuscript NCOMMS-18-07950-T

The manuscript from Wu et al. aims at revealing a new mechanism underlying the organ shape morphological trait in plants. In tomato, the shape of the fruit is determined by the OVATE gene, which belongs to the multigene family encoding the OVATE Family Proteins (OFP). In Arabidopsis and rice, the shapes of a leaf and grain are determined by LONGIFOLIA1 and LONGIFOLIA2, which belongs to the TONNEAU1 Recruiting Motif (TRM) family. In this report, the authors established a link between OFPs and TRMs in the developmental pathway that governs the shape of tomato fruit.

First, they fine-mapped and cloned an ovate-phenotype shape modifier locus (named *sov1*) harbouring an OFP encoding gene, designed thereafter as SIOFP20. By overexpressing SIOFP20 under the 35S promoter in the yellow pear tomato genotype, they confirmed the observed effect associated to *sov1*: the overexpression of SIOFP20 produced round fruits, and at the vegetative state, round leaflets and shortened leaves. When SIOFP20 was suppressed in the round fruit-shape *S. pimpinellifolium* species, elongated fruits were obtained but only in an ovate background. Second the authors looked for protein interactors of OVATE using yeast two-hybrid screens, for which a vast majority out of 185 interactors belonged to the TRM multigene family. Several of these TRMs interacting with OVATE were found to interact also with SIOFP20. These interactions were characterized at the aminoacid sequence levels, but more importantly confirmed in planta, using BiFC. Subcellular localization assays co-expressing the protein partners showed that the interaction between specific OFPs and TRMs induces a relocalization of the proteins within different cellular compartments, such as TRMs associated to microtubules are relocated to cytoplasm by OFPs, and OFPs are recruited to microtubules by TRMs.

Next the authors have studied the effects of OFPs and TRMs on fruit shape by creating near-isogenic lines for the natural ovate and *sov1* alleles, and generated mutated versions of TRMs using the CRISPR-Cas9 technology. Interestingly they showed that the *Sltrm5* mutation can complement the ovate/*sov1* background, giving rise to round-shaped fruits. An in-depth histological characterization of the mutants in anthesis ovaries showed that different cellular patterns were observed in relationship to the shape of the ovary in its proximal end. Cell number increased along the proximo-distal axis and decreased along the medio-lateral axis in the elongated ovaries of ovate and ovate/*sov1*. In accordance with its ability to complement the ovate/*sov1* phenotype, the *trm5* mutation introgressed ovate/*sov1* resumed the wild-type phenotype.

Finally, in both cucumber and potato, the authors showed nice evidence that OFPs operate in a conserved pathway to determine fruit and tuber shape. In addition, they fine-mapped the fruit shape QTL *fs2.1* in cucumber, and found it is associated to an ortholog of SITRM5.

The findings described herein are quite original and of a great importance for the plant development field. The discovery in this original paper clearly points at a protein pathway that determines the shape of plant organs, via a mechanism operating at the level of cell division and cell growth. However with regard to the discovery of TRMs interacting with OFPs, and this new mechanism, the manuscript stays somewhat descriptive as much as the cellular context is concerned. It would have been nice to get mechanistic insights about the potential role of these associating proteins in either promoting cell divisions, or influencing in a way the orientation of cell division plates to favor one direction for fruit growth (elongated ovaries?). For instance, from the ovary cross-sections (performed for figure 3), a more detailed analysis related to planes of division, as putatively influenced by the mutation of TRM5 would be worth investigating or at least also mentioning if significant.

As a general comment, the methodology used in this manuscript is fully relevant, and the authors provided a very detailed methods description, enabling any researcher to reproduce the work. Care should be given to clarify the method of cell number determination as questioned in the above paragraph. The data are quite nicely presented in five figures, although the density of

provided information in each figure is already quite high. The authors performed all statistical analyses very appropriately to strengthen the conclusions. Additional data really useful for the understanding are provided as supplemental figures.

Specific questions and comments the authors may find useful for improving a revised manuscript:

- Figure 1c, 1d, and figure 3a: I would suggest show the fruit sections upside down, as to put the columella down (proximal end of the fruit), to be easily understandable for the reader relative to the ovary characterization shown in figure 3c, d, e. Or the other way around, if the showing the pear-shaped fruit in this position is the most important.
- Figure 2c: correct the above title "Interactions between SIOFPs and SITRMs" into "Interactions between SIOFPs and SITRMs".
- Page 5: the 35S promoter was used to overexpress SIOFP20. Clearly fruit- and leaf shapes are affected, but what happens to the plant vegetative development? Is there any effect on plant size for instance? Please comment.
- Page 9: lines 185-188, it is tempting to foresee the effects of the interaction between OFP and TRMs. However, this sentence at this stage sounds quite speculative in the absence of clear demonstration.
- Page 9: as SITRM5 was shown in this manuscript to have clear complementation effects on fruit shape when mutated in the ovate/sov1 background, it would be worth investigating whether SITRM5 is expressed in a cell cycle manner, during the M phase, using tobacco BY-2 synchronized cells.
- Page 9: the authors do not comment the vegetative phenotype for the ami-RNA trm mutants. Are there any effects on plant development? When compared to what is known in Arabidopsis for these mutants, is there any phenotype at the cellular level, in terms of modifications in the cell division plate orientation?
- Page 10 - Figure 3c, d, e, and Methods section (page 26; lines 653-657): the way cell number was determined in the proximal area of the ovary needs clarification for the readers. Was the counting of cells done in the rectangular surface drawn on figure 3d? If yes, it means it was a general counting including cells forming the basis of the columella, as well as cells belonging to the ovary wall? Is this sound to consider these two tissues and herewith constituting cells as "behaving" likely?
- Page 11, lines 224-225: if "modulating cell division patterns" refers to a stimulation in cell division, as indicated from the morphological changes (number of cells), it would be worth giving some data about cell division activities per se, by counting mitotic figures in ovary cross sections after DAPI staining or using another dye.
- Figure 3e: all graphs lack of a y-axis name; Relative value alike Figure 3b, c?
- Figure 3e: the authors provided cellular characteristics, such as cell length and cell width, but do not comment on cell area per se. Clearly, in ovate and ovate/sov1 cells are bigger, and cell expansion together with cell division contribute as the final size and shape of the fruit, and are thus influenced as early as at anthesis in these lines.

Reviewer #2 (Remarks to the Author):

The manuscript entitled "A novel mechanism underlies morphological diversity in plants" by Wu S. and collaborators presents novel and very interesting insights about the regulation of fruit and in general aerial organ shape, providing explanations on the mechanism underlying differences in organ morphology among tomato cultivars. The manuscript demonstrates by using genetic approaches and molecular analyses, that OVATE and SIOFP20 are key factors in regulating fruit morphology in tomato. An original and novel finding is the demonstration that OVATE and SIOFP20 physically interact with proteins of the TRM family. Members of TMR family was previously implicated in the control of aerial part and grain shape in Arabidopsis and rice, respectively. The interaction between SIOFP and TMR appears crucial for modulating the cell division pattern in the tomato ovaries at anthesis. The manuscript provides evidence that orthologous OFP and TRM

genes can be involved in controlling organ morphology in other crop species.

In my opinion, the data presented in the manuscript represent a significant improvement of the present knowledge on the molecular mechanisms regulating fruit shape and are relevant for the scientific community working in the field. However, I think that the manuscript needs some revisions prior to publication, in particular concerning the organization and presentation of the results and the discussion of the mechanism of action of OVATE/SIOFP20/TRM5 complex.

My suggestions are:

Title

The title should describe more precisely the content of the manuscript. In my opinion, the present title is rather generic.

Introduction:

I think that more details on the physiological role and molecular function of OVATE/OFP and TRM family should be included.

Results

Some data seem redundant, the authors should check for repetitions in the figures present in the text and the supplementary ones. For instance, the M8 motif is reported twice, also some YH2 data seems somewhat repeated. I suggest in general to organize better the supplementary material avoiding reiteration and proceeding in a logical sequence. For instance, the panel B of Supplementary Fig.1 is anticipated with respect to the description of the results in the text.

The part concerning the experiments on the interaction between OVATE and SIOFP20 with different members of the TRM family is quite confusing (Fig2 and Supplementary Fig7 and 8). It is not clear why some experiments of expression and co-expression in *N. benthamiana* leaf epidermal cells and BiFC are carried out with some TRM proteins and not with others. I found this part of the manuscript quite difficult to follow and confusing. I would suggest presenting the data consistently for TRM5 and TRM3/4, which are the two TRM chosen for mutation and complementation experiments and moving the other data to the supplementary material.

My criticisms concerns also the emphasis on the co-expression experiments that in the author view support the evidence that the physical interactions between the members of these two families led to dynamic re-localization of the protein complexes to different cellular compartments. I think that the co-expression does not give a direct proof of the interaction of the proteins and the re-localization of the complexes. In addition, it seems that the authors used in these expression experiments the cds of the genes under the 35S promoter and not the entire genes with their regulatory regions.

The BiFC representing an independent demonstration of the in vivo interaction of these proteins is not well described; for instance, it is not explained where the interaction takes place in the cell. In addition, BiFC has been carried out only for OVATE and TMR5 and not for SIOFP20 and TMR5. The BiFC experiment on OVATE and TMR 17/20 can be omitted. The BiFC data should be improved. I'd suggest to improve the discussion on how the formation of the SIOFP/TMR complexes could affect the cell division pattern in the ovary.

I think that besides the analysis of the expression pattern of OVATE/SIOFP20 and TRM5 transcript during different phases of ovary/fruit development, it would be interesting to determine the changes in proteins' abundance to support the possibility of interaction of these proteins during the initial phases of ovary growth.

Statistical analysis is appropriate. The methods are properly described.

Reviewer #3 (Remarks to the Author):

The manuscript describes the mapping of a locus involved in fruit shape referred to as suppressor of ovate (*sov1*). The study identified a 149.7-kb region on chromosome 10 carrying three annotated genes, *Solyc10g076170*, *Solyc10g076180* and *Solyc10g076190* among which *Solyc10g076180*, corresponding to a member of the ovate family of proteins (OFP) - here named

SIOFP20, was differentially expressed in near isogenic lines polymorphic at the *sov1* locus. Analyzing allelic diversity at *sov1* locus showed a 31-kb deletion residing 6.5 kb upstream of the transcription start site of SIOFP20. The authors made the hypothesis that this deletion may result in the reduced expression SIOFP20 gene. While Ovate mutation leads to elongated fruit shape, *sov1* mutation suppresses the elongated phenotype presumably by reducing OFP20 expression thus resulting in round shape. Reverse genetics approaches via over- and under-expression of this gene in the appropriate genetic background validated its ability to control fruit shape. The plant transformation data support the idea that mutations in two OFP members contribute to natural fruit shape variation in the tomato germplasm. In their main conclusion, the authors state that cell division rather than cell elongation/expansion accounts for the elongated fruit shape and that OFP and TRM emerge as key players in controlling this process even though how these genes/proteins impact this morphological trait remains unexplained. While overall, the study brings some convincing evidences that SIOFP20 is involved in shape determination through analyzing genetic diversity in tomato and other species, however some issues still need to be solved in order to make the story complete.

The following issues need to be addressed:

- 1- It is rather surprising, that the deletion affects the expression of OFP20 (Solyc1010g07180) but not that of Solyc10g076170 located closer to the mutation site. In this regard, because the expression of Solyc10g076170 has been tested only in flower anthesis, it cannot be ruled out that the mutation may affect its expression in other tissues or organs. Therefore, the statement that the mutation doesn't affect its expression needs to be tuned down unless they assess its expression level in other tissues. Regarding the phenotype description of the *sov1* mutation, it would be interesting to know whether they also affect flower, fruit set and vegetative growth like hypocotyl, roots, cotyledons, since OFP20 is highly expressed in these tissues.
- 2- OFP8 expression almost perfectly mimics that of OFP20 (heatmap Sup Fig1b). May these two OFP family members have redundant function?
- 3- Why Figure 1C shows only T0 transformants? Is it because the phenotype of overexpressing lines is unstable through generation?
- 4- Why the non-targeted Y-2-H screen was made with Ovate and not with OFP20?
- 5- The selection of candidates for the co-expression experiments doesn't seem logical. Indeed, among 11 TRM identified by Y-2-H, only 5 displayed expression pattern similar to Ovate. It is not clear what motivated the selection of TRM3/4 and 17/20a for transient co-expression since the expression of these TRM members is divergent with that of Ovate and OFP20 in all tissues tested (see heatmap Fig1b). It is not true that the expression of SITRM5 and SITRM25 follows the same dynamics than Ovate. Based on the heatmap (Supplementary Figure 1), there are better candidates like SITRM19, SITRM22, SITRM6/7/8a and SITRM6/7/8b. Also, TRM 16/32a, TRM 16/32b and TRM1/2/3/4/5 are better candidate interesting candidates as they are co-expressed with OFP20 at flower anthesis.
- 6- Leaf tissues is apparently not the most appropriate to test the subcellular localization as both ovate and OFP20 exhibit extremely low expression level in this organ (see Sup Fig1a).
- 7- The interpretation of Figure 2j and 2k can somehow be questionable. SITRM5 doesn't seem to really relocate to the cytoplasm when co-expressed with OVATE, and the latter doesn't remain in the cytoplasm but rather spread on the entire cell surface (Fig. 2j). Likewise, when co-expressed with SIOFP20, SITRM5 doesn't show the same association with the microtubule as when it is expressed alone (Fig. 2k).
- 8- For better clarity, Supplementary Figure 7 should also include fluorescence microscopy pictures.
- 9- Establishing a link between the subcellular localization of OFP/TRM protein complexes and cell division determinism is an overstatement as it is not supported by direct experimental evidence.
- 10- How do Ovate and OFP20 proteins interact or cooperate?
- 11- A simplified scheme depicting the regulation mode of OFP/TRM proposed would be useful to illustrate the mechanism uncovered in the present study.
- 12- In the Abstract, the authors claim that OFP/TRM interact in a novel pathway altering fruit shape, but these genes and the corresponding encoded proteins have been already associated to fruit shape.

13- Deep transcriptional profiling of early phases of fruit development is missing as it may give clues on the gene regulatory networks or signaling pathways involved. The authors have the relevant tomato genetic material with dramatic contrast regarding the studied trait, so it is advised that they perform this comparative genome-wide transcriptomic profiling.

Response to reviewers' comments:

Reviewer #1 (Remarks to the Author):

Manuscript NCOMMS-18-07950-T

The manuscript from Wu et al. aims at revealing a new mechanism underlying the organ shape morphological trait in plants. In tomato, the shape of the fruit is determined by the OVATE gene, which belongs to the multigene family encoding the OVATE Family Proteins (OFP). In Arabidopsis and rice, the shapes of a leaf and grain are determined by LONGIFOLIA1 and LONGIFOLIA2, which belongs to the TONNEAU1 Recruiting Motif (TRM) family. In this report, the authors established a link between OFPs and TRMs in the developmental pathway that governs the shape of tomato fruit.

First, they fine-mapped and cloned an ovate-phenotype shape modifier locus (named *sov1*) harbouring an OFP encoding gene, designed thereafter as SIOFP20. By overexpressing SIOFP20 under the 35S promoter in the yellow pear tomato genotype, they confirmed the observed effect associated to *sov1*: the overexpression of SIOFP20 produced round fruits, and at the vegetative state, round leaflets and shortened leaves. When SIOFP20 was suppressed in the round fruit-shape *S. pimpinellifolium* species, elongated fruits were obtained but only in an ovate background.

Second the authors looked for protein interactors of OVATE using yeast two-hybrid screens, for which a vast majority out of 185 interactors belonged to the TRM multigene family. Several of these TRMs interacting with OVATE were found to interact also with SIOFP20. These interactions were characterized at the amino acid sequence levels, but more importantly confirmed in planta, using BiFC. Subcellular localization assays co-expressing the protein partners showed that the interaction between specific OFPs and TRMs induces a relocation of the proteins within different cellular compartments, such as TRMs associated to microtubules are relocated to cytoplasm by OFPs, and OFPs are recruited to microtubules by TRMs.

Next the authors have studied the effects of OFPs and TRMs on fruit shape by creating near-isogenic lines for the natural ovate and *sov1* alleles, and generated mutated versions of TRMs using the CRISPR-Cas9 technology. Interestingly they showed that the *Sltrm5* mutation can complement the ovate/*sov1* background, giving rise to round-shaped fruits. An in-depth histological characterization of the mutants in anthesis ovaries showed that different cellular patterns were observed in relationship to the shape of the ovary in its proximal end. Cell number increased along the proximo-distal axis and decreased along the medio-lateral axis in the elongated ovaries of ovate and ovate/*sov1*. In accordance with its ability to complement the ovate/*sov1* phenotype, the *trm5* mutation introgressed ovate/*sov1* resumed the wild-type phenotype.

Finally, in both cucumber and potato, the authors showed nice evidence that OFPs operate in a conserved pathway to determine fruit and tuber shape. In addition, they fine-mapped the fruit shape QTL *fs2.1* in cucumber, and found it is associated to an ortholog of SITRM5.

The findings described herein are quite original and of a great importance for the plant development field. The discovery in this original paper clearly points at a protein pathway that determines the shape of plant organs, via a mechanism operating at the level of cell division and cell growth. However with regard to the discovery of TRMs interacting with OFPs, and this new mechanism, the manuscript stays somewhat descriptive as much as the cellular context is concerned. It would have been nice to get mechanistic insights about the potential role of these associating proteins in either promoting cell divisions, or influencing in a way the orientation of cell division plates to favor one direction for fruit growth (elongated ovaries?). For instance, from the ovary cross-sections (performed for figure 3), a more detailed analysis related to planes of

division, as putatively influenced by the mutation of TRM5 would be worth investigating or at least also mentioning if significant.

As a general comment, the methodology used in this manuscript is fully relevant, and the authors provided a very detailed methods description, enabling any researcher to reproduce the work. Care should be given to clarify the method of cell number determination as questioned in the above paragraph. The data are quite nicely presented in five figures, although the density of provided information in each figure is already quite high. The authors performed all statistical analyses very appropriately appropriate to strengthen the conclusions. Additional data really useful for the understanding are provided as supplemental figures.

RESPONSE: We thank the reviewer for the encouraging comments. We agree that mechanistic insights would be helpful. However, identifying cell division planes requires the identification of when these changes in cell division planes occur in the developing organ. The images shown in figure 3 (new figure 4) would not give useful insights into division planes since the ovary at anthesis already shows the shape changes. Thus, the cell division plane changes would have occurred earlier in development. As of now, we don't have that information.

Specific questions and comments the authors may find useful for improving a revised manuscript:

- Figure 1c, 1d, and figure 3a: I would suggest show the fruit sections upside down, as to put the columella down (proximal end of the fruit), to be easily understandable for the reader relative to the ovary characterization shown in figure 3c, d, e. Or the other way around, if the showing the pear-shaped fruit in this position is the most important.

RESPONSE: Thanks for the suggestion. The ovary sections have been rotated in Figure 3 (new figure 4).

- Figure 2c: correct the above title "Interactions between SIOFPs and SITRMs" into "Interactions between SIOFPs and SITRMs".

RESPONSE: The misspelling has been corrected.

- Page 5: the 35S promoter was used to overexpress SIOFP20. Clearly fruit- and leaf shapes are affected, but what happens to the plant vegetative development? Is there any effect on plant size for instance? Please comment.

RESPONSE: We did not extensively analyze vegetative development because our focus was on fruit shape. The leaf shape was noticeably altered which is why we measured and reported it.

- Page 9: lines 185-188, it is tempting to foresee the effects of the interaction between OFP and TRMs. However, this sentence at this stage sounds quite speculative in the absence of clear demonstration.

RESPONSE: we agree with the reviewer and removed the speculative sentence.

- Page 9: as SITRM5 was shown in this manuscript to have clear complementation effects on fruit shape when mutated in the ovate/sov1 background, it would be worth investigating whether SITRM5 is expressed in a cell cycle manner, during the M phase, using tobacco BY-2 synchronized cells.

RESPONSE: Indeed, it would be helpful to know whether any of the TRMs are expressed in a cell cycle-specific manner. We don't have such a resource for tomato and expression in BY-2 cells might not be informative because it is a heterologous system.

- Page 9: the authors do not comment the vegetative phenotype for the ami-RNA trm mutants. Are there any effects on plant development? When compared to what is known in Arabidopsis for

these mutants, is there any phenotype at the cellular level, in terms of modifications in the cell division plane orientation?

RESPONSE: The T-DNA insertion mutant of *AtOFPI* has no phenotype (Wang et al., 2011. *PLoS One*) whereas knock outs or overexpressors of *AtTRM5* have not been reported. Note however, that Arabidopsis and many Brassicaceae that we checked harbor a TRM5 gene without an M8 (ovate-interacting motif). This data is not shown in the manuscript. Regardless, all TRM5 genes outside the Brassicaceae family (cucumber, tomato, and many others) carry M8. Thus, whereas the role of *AtTRM5* is unclear, its role in other species seems clear. The role of the tomato TRM5 in Arabidopsis might be represented by TRM1 and TRM2, which could make these TRMs more likely to be orthologous to each another. The roles *AtTRM1* and *AtTRM2* (*LONGIFOLIA2* and *LONGIFOLIA1*) are represented by the activation-tagging lines and T-DNA insertion mutations (Lee et al., 2006. *Development*). Overexpression of *AtTRM1* or *AtTRM2* promotes cell elongation, leading to longer lateral organs in the aerial part of the plant; while the double mutant has decreased leaf length as a result of less elongated cells. In rice however, the role of the TRM5 ortholog on grain shape has been attributed to both cell length and cell number (Wang et al, 2015. *Nat Genet*; Wang et al, 2015. *Nat Genet*; Zhou et al, 2015. *Genetics*). Thus defects in these TRMs are either associated with cell shape and/or cell number which is consistent with our findings in tomato. In tomato, KO TRM5 also has a clear leaf shape phenotype (Supplementary figure 11) as seen in the loss-of-function mutants of *AtTRM1* and *AtTRM2* in Arabidopsis. We decided to underreport this defect because the focus of the tomato part of the study is on fruit shape. No other obvious vegetative phenotypes are found in the tomato TRM5 KO. There is no information from Arabidopsis on cell division plane orientation.

- Page 10 - Figure 3c, d, e, and Methods section (page 26; lines 653-657): the way cell number was determined in the proximal area of the ovary needs clarification for the readers. Was the counting of cells done in the rectangular surface drawn on figure 3d? If yes, it means it was a general counting including cells forming the basis of the coumella, as well as cells belonging to the ovary wall? Is this sound to consider these two tissues and herewith constituting cells as “behaving” likely?

RESPONSE: Yes, the counting of the cells was done in the rectangular area because that area seems the most affected in *ovate/sov1*. We edited in the methods section that only the parenchyma cells were counted and no other cell types.

- Page 11, lines 224-225: if “modulating cell division patterns” refers to a stimulation in cell division, as indicated from the morphological changes (number of cells), it would be worth giving some data about cell division activities per se, by counting mitotic figures in ovary cross sections after DAPI staining or using another dye.

RESPONSE: We used the term “modulating” deliberately so as to not pin ourselves down to stimulation of cell division, in one direction only, or another more complex situation. As we addressed above, we don’t know yet when changes in ovary shape become apparent and therefore have not evaluated the mitotic figures in ovary longitudinal sections. Anthesis stage ovaries are considered stationary and not dividing much, until the signal of pollination and fertilization leads to a rapid increase in cell division (Gillaspy et al, 1993; Xiao et al, 2009). This is why the evaluation of cell division at this stage will unlikely be informative.

- Figure 3e: all graphs lack of a y-axis name; Relative value alike Figure 3b, c?

RESPONSE: Thanks for pointing this out. We’ve added the y-axis names.

- Figure 3e: the authors provided cellular characteristics, such as cell length and cell width, but do not comment on cell area per se. Clearly, in ovate and ovate/sov1 cells are bigger, and cell

expansion together with cell division contribute as the final size and shape of the fruit, and are thus influenced as early as at anthesis in these lines.

RESPONSE: We agree with the reviewer that cell size is changed, which is pointed out in the text that the *ovate* and *ovate/sov1* mutants have increased cell length and cell width. However, in the context of organ shape, the cell length and width increases are highly variable and not consistent with a markedly enough change in cell shape. Specifically, cell shape in *ov/sov1* is not significantly different from wild type.

Reviewer #2 (Remarks to the Author):

The manuscript entitled “A novel mechanism underlies morphological diversity in plants” by Wu S. and collaborators presents novel and very interesting insights about the regulation of fruit and in general aerial organ shape, providing explanations on the mechanism underlying differences in organ morphology among tomato cultivars. The manuscript demonstrates by using genetic approaches and molecular analyses, that OVATE and SIOFP20 are key factors in regulating fruit morphology in tomato. An original and novel finding is the demonstration that OVATE and SIOFP20 physically interact with proteins of the TRM family. Members of TMR family was previously implicated in the control of aerial part and grain shape in Arabidopsis and rice, respectively. The interaction between SIOFP and TMR appears crucial for modulating the cell division pattern in the tomato ovaries at anthesis. The manuscript provides evidence that orthologous OFP and TRM genes can be involved in controlling organ morphology in other crop species.

In my opinion, the data presented in the manuscript represent a significant improvement of the present knowledge on the molecular mechanisms regulating fruit shape and are relevant for the scientific community working in the field. However, I think that the manuscript needs some revisions prior to publication, in particular concerning the organization and presentation of the results and the discussion of the mechanism of action of OVATE/SIOFP20/TRM5 complex.

My suggestions are:

Title

The title should describe more precisely the content of the manuscript. In my opinion, the present title is rather generic.

RESPONSE: First, we thank the reviewer for the positive response on the manuscript. Regarding the title, we prefer the generic title because it should entice and attract readers outside the field.

Introduction:

I think that more details on the physiological role and molecular function of OVATE/OFP and TRM family should be included.

RESPONSE: In this manuscript, we discover the most likely molecular function of OFPs, which is to move protein complexes at the subcellular level by using/changing cytoskeleton activities. In our opinion, the current literature about OFPs and their biochemical function (transcriptional repressors) is fundamentally wrong. Thus, the main reason why we kept the introduction about OFP short is not to have to deal with the misinformation about its function up front. Regarding TRMs, we think the story is presented better at a later stage in the manuscript where we describe in more detail than in other publications the cellular basis of mutations in one of the TRMs.

Results

Some data seem redundant, the authors should check for repetitions in the figures present in the text and the supplementary ones. For instance, the M8 motif is reported twice, also some YH2 data seems somewhat repeated. I suggest in general to organize better the supplementary material

avoiding reiteration and proceeding in a logical sequence. For instance, the panel B of Supplementary Fig.1 is anticipated with respect to the description of the results in the text.

RESPONSE: Thanks for pointing this out. We agree with the reviewer and removed redundancy throughout the text and supplementary figures.

The part concerning the experiments on the interaction between OVATE and SIOFP20 with different members of the TRM family is quite confusing (Fig2 and Supplementary Fig7 and 8). It is not clear why some experiments of expression and co-expression in *N. benthamiana* leaf epidermal cells and BiFC are carried out with some TRM proteins and not with others. I found this part of the manuscript quite difficult to follow and confusing. I would suggest presenting the data consistently for TRM5 and TRM3/4, which are the two TRM chosen for mutation and complementation experiments and moving the other data to the supplementary material.

RESPONSE: We agree with the reviewer. Part of the issue was that many experiments started well before we knew the relevance of TRM5. The other issue was that some OFFP-TRM interactions worked much better in one system than the other. This effectively meant that some OFFP-TRM combinations were tested much more extensively than others, and that depended on what system we used (tobacco or yeast). We revised the text to make this section flow better and more focused on the TRM5. We added additional BiFC data including the subcellular localization of the interactions.

My criticisms concerns also the emphasis on the co-expression experiments that in the author view support the evidence that the physical interactions between the members of these two families led to dynamic re-localization of the protein complexes to different cellular compartments. I think that the co-expression does not give a direct proof of the interaction of the proteins and the re-localization of the complexes.

RESPONSE: We respectfully disagree with this statement because we demonstrate that mutations in the interaction motifs lead to less relocalization. While it is true that this is still indirect, it is hard to argue for another explanation than that they relocalize upon interaction: we abolish interaction through mutations and the proteins relocalize less or not at all. However, to further confirm, we have since conducted BiFC with OFFP20-TRM5 and show that, as expected, they interact while associated with the microtubules. We show the same for OVATE-TRM5 and that they interact in the cytosol, which is also as expected. This data can be found in the new Figure 3.

In addition, it seems that the authors used in in these expression experiments the cds of the genes under the 35S promoter and not the entire genes with their regulatory regions.

RESPONSE: Correct. Everyone uses the 35S promoter for the expression analyses in tobacco. The endogenous promoter is likely not expressing in mature leaf cells (see expression levels of OVATE and TRM5 in mature leaves in Supplementary fig. 1). Thus, using endogenous promoters would not yield any usable data. Moreover, it is unclear whether the tomato TRM or OFFP promoters work in tobacco.

The BiFC representing an independent demonstration of the in vivo interaction of these proteins is not well described; for instance, it is not explained where the interaction takes place in the cell. In addition, BiFC has been carried out only for OVATE and TMR5 and not for SIOFP20 and TMR5. The BiFC experiment on OVATE and TMR 17/20 can be omitted. The BiFC data should be improved.

RESPONSE: We included additional BiFC data as well as an overview and with mutant versions of the proteins and a close view for subcellular visualization. See comments above.

I'd suggest to improve the discussion on how the formation of the SIOFP/TMR complexes could affect the cell division pattern in the ovary.

RESPONSE: Thanks for the suggestion. We added a model describing the interaction based on the evidence we show in the paper. The model describes how expression levels of OFPs and TRMs affects shape in organs in many of not all higher plants.

I think that besides the analysis of the expression pattern of OVATE/SIOFP20 and TRM5 transcript during different phases of ovary/fruit development, it would be interesting to determine the changes in proteins' abundance to support the possibility of interaction of these proteins during the initial phases of ovary growth.

RESPONSE: We agree it would be nice to have additional supportive data to demonstrate the likelihood of interactions in vivo. However, we have no antibodies against the proteins and creating fusion constructs in plants are laborious. This is primarily because of the fact that the genes would need to be expressed at endogenous levels to obtain interpretable information.

Statistical analysis is appropriate. The methods are properly described.

Reviewer #3 (Remarks to the Author):

The manuscript describes the mapping of a locus involved in fruit shape referred to as suppressor of ovate (*sov1*). The study identified a 149.7-kb region on chromosome 10 carrying three annotated genes, *Solyc10g076170*, *Solyc10g076180* and *Solyc10g076190* among which *Solyc10g076180*, corresponding to a member of the ovate family of proteins (OFP) - here named SIOFP20, was differentially expressed in near isogenic lines polymorphic at the *sov1* locus. Analyzing allelic diversity at *sov1* locus showed a 31-kb deletion residing 6.5 kb upstream of the transcription start site of SIOFP20. The authors made the hypothesis that this deletion may result in the reduced expression SIOFP20 gene. While Ovate mutation leads to elongated fruit shape, *sov1* mutation suppresses the elongated phenotype presumably by reducing OFP20 expression thus resulting in round shape. Reverse genetics approaches via over- and under-expression of this gene in the appropriate genetic background validated its ability to control fruit shape. The plant transformation data support the idea that mutations in two OFP members contribute to natural fruit shape variation in the tomato germplasm. In their main conclusion, the authors state that cell division rather than cell elongation/expansion accounts for the elongated fruit shape and that OFP and TRM emerge as key players in controlling this process even though how these genes/proteins impact this morphological trait remains unexplained. While overall, the study brings some convincing evidences that SIOFP20 is involved in shape determination through analyzing genetic diversity in tomato and other species, however some issues still need to be solved in order to make the story complete.

The following issues need to be addressed:

1- It is rather surprising, that the deletion affects the expression of OFP20 (*Solyc10g076180*) but not that of *Solyc10g076170* located closer to the mutation site. In this regard, because the expression of *Solyc10g076170* has been tested only in flower anthesis, it cannot be ruled out that the mutation may affect its expression in other tissues or organs. Therefore, the statement that the mutation doesn't affect its expression needs to be tuned down unless they assess its expression level in other tissues. Regarding the phenotype description of the *sov1* mutation, it would be interesting to know whether they also affect flower, fruit set and vegetative growth like hypocotyl, roots, cotyledons, since OFP20 is highly expressed in these tissues.

RESPONSE: We thank the reviewer for the comments and positive response. The reviewer is correct that we did not show the expression of *Solyc10g076170* in many tissues. In an effort to focus the manuscript, we honed in on OFP20 right away without spending much time on the other candidate genes because OFP20 was a very plausible candidate gene. To come back to the unknown gene that is in the deleted segment, a quick search in the tomato functional genomics database shows that *Solyc10g076170* is not expressed or at extremely low levels (<http://ted.bti.cornell.edu/cgi-bin/TFGD/digital/home.cgi>). In addition, *Solyc10g076170* is not likely to function as a protein-coding gene because the putative protein only corresponds to a single protein with partial homology to a pepper hypothetical mitochondrial protein. We did not evaluate fruit set and other reproductive as well as vegetative traits because the main focus was on the shape of tomato fruits. We added a sentence to that part of the manuscript to indicate more clearly why we only pursued OFP20.

2- OFP8 expression almost perfectly mimics that of OFP20 (heatmap Sup Fig1b). May these two OFP family members have redundant function?

RESPONSE: We have not tested redundancy between these two OFPs. Also, OFP20 and OFP8 expression patterns (very high at anthesis while the shape is already determined) do not explain how these OFPs might affect organ shape. This is why understanding the role of OFP8 might not be informative in the context of organ shape.

3- Why Figure 1C shows only T0 transformants? Is it because the phenotype of overexpressing lines is unstable through generation?

RESPONSE: The phenotype is stable across generations. We have tested the T1 using two primary T0 lines and the results are in the supplementary figure 3.

4- Why the non-targeted Y-2-H screen was made with Ovate and not with OFP20?

RESPONSE: The Y2H started with OVATE because OFP20 had not been mapped yet. After we noticed that OFP20 was a likely candidate for *sov1*, we started experiments to show that OVATE-interacting TRMs also interact with OFP20.

5- The selection of candidates for the co-expression experiments doesn't seem logical. Indeed, among 11 TRM identified by Y-2-H, only 5 displayed expression pattern similar to Ovate. It is not clear what motivated the selection of TRM3/4 and 17/20a for transient co-expression since the expression of these TRM members is divergent with that of Ovate and OFP20 in all tissues tested (see heatmap Fig1b). It is not true that the expression of SITRM5 and SITRM25 follows the same dynamics than Ovate. Based on the heatmap (Supplementary Figure 1), there are better candidates like SITRM19, SITRM22, SITRM6/7/8a and SITRM6/7/8b. Also, TRM 16/32a, TRM 16/32b and TRM1/2/3/4/5 are better candidate interesting candidates as they are co-expressed with OFP20 at flower anthesis.

RESPONSE: As to the concerns raised by reviewer 2 and 3, our presentation of the data in the submitted manuscript led to confusion. We revised the text and figures accordingly to focus more on TRM5 and less on other TRMs, especially for the experiments involving tobacco. In addition, many of the TRMs reviewer 3 mentions were not recovered in the Y2H screen because they lack the M8 motif.

6- Leaf tissues is apparently not the most appropriate to test the subcellular localization as both ovate and OFP20 exhibit extremely low expression level in this organ (see Sup Fig1a).

RESPONSE: the reviewer is correct. However, the *N. benthamiana* coexpression studies are used extensively to address protein interactions irrespective of biological relevance of the system. It is the first step prior to identifying these interactions in a relevant but experimentally more challenging system.

7- The interpretation of Figure 2j and 2k can somehow be questionable. SITRM5 doesn't seem to really relocate to the cytoplasm when co-expressed with OVATE, and the latter doesn't remain in the cytoplasm but rather spread on the entire cell surface (Fig. 2j). Likewise, when co-expressed with SIOFP20, SITRM5 doesn't show the same association with the microtubule as when it is expressed alone (Fig. 2k).

RESPONSE: To show microtubule association, you need to show a Z-stack. To show cytoplasmic or nuclear localization, you need to show a single section. We think the reviewer is inquiring about a Z-stack as well as individual section image for OVATE and OFP20, and we have added those (new Fig. 3a-b). We hope that will alleviate the concerns the reviewer raised. When co-expressing OVATE with SITRM5, we clearly see the cytoplasm localization of SITRM5 (Fig. 3e) similar to that of OVATE when expressed alone (Fig. 3a). When co-expressing SIOFP20 with SITRM5, SIOFP20 decorates the microtubules (Fig. 3f), which is a pattern not seen when SIOFP20 is expressed alone (Fig. 3b).

8- For better clarity, Supplementary Figure 7 should also include fluorescence microscopy pictures.

RESPONSE: While we agree with the reviewer, the situation is somewhat fluid especially with the number of categories (3 for OFP20-TRM5 and 2 for OVATE-TRM5) times the number of mutations. Thus, we would have to show many images neither of which will be highly informative. The findings that we aim to present is the counting of the cells that show a certain co-localization pattern, which is what is shown in Supplementary figure 7.

9- Establishing a link between the subcellular localization of OFP/TRM protein complexes and cell division determinism is an overstatement as it is not supported by direct experimental evidence.

RESPONSE: We agree with the reviewer and deleted the statement.

10- How do Ovate and OFP20 proteins interact or cooperate?

RESPONSE: There is no evidence of protein interaction between the OFPs. In addition, the expression dynamics differ quite a bit. We think that OVATE sets up the pattern of cell division and slightly later, OFP20 enhances the pattern. Extension like a telescope.

11- A simplified scheme depicting the regulation mode of OFP/TRM proposed would be useful to illustrate the mechanism uncovered in the present study.

RESPONSE: We thank the reviewer for the suggestion. We have added the model (new figure 7) for interaction that is supported by the evidence we have gathered in this manuscript.

12- In the Abstract, the authors claim that OFP/TRM interact in a novel pathway altering fruit shape, but these genes and the corresponding encoded proteins have been already associated to fruit shape.

RESPONSE: Correct, but the OFP-TRM interaction is novel and has not been described before.

13- Deep transcriptional profiling of early phases of fruit development is missing as it may give clues on the gene regulatory networks or signaling pathways involved. The authors have the relevant tomato genetic material with dramatic contrast regarding the studied trait, so it is advised that they perform this comparative genome-wide transcriptomic profiling.

RESPONSE: We thank the reviewer for this suggestion. However, this experiment would be an excellent start of a follow up manuscript.

Reviewers' comments:

Reviewer #1 (Remarks to the Author):

Manuscript NCOMMS-18-07950A

"A novel mechanism underlies morphological diversity in plants" by Wu et al.

This manuscript from Wu et al. is a resubmission of a previously reviewed manuscript. It describes a new mechanism underlying the organ shape morphological trait in plants, especially in tomato where the shape of the fruit is determined by the OVATE gene. OVATE belongs to the multigene family encoding the OVATE Family Proteins (OFP). In Arabidopsis and rice, the shapes of a leaf and grain are determined by LONGIFOLIA1 and LONGIFOLIA2, which belongs to the TONNEAU1 Recruiting Motif (TRM) family. In this report, the authors established a link between OFPs and TRMs in the developmental pathway that governs the shape of tomato fruit.

In my previous review I pointed out that the manuscript was somewhat descriptive and that some more information regarding the cellular context would be expected as to go into mechanistic insights about the potential role of these associating proteins in either promoting cell divisions, or influencing in a way the orientation of cell division plates to favor one direction for fruit growth. I acknowledge that this question is rather difficult to address in developing tomato fruits and it may go far beyond the necessary and reasonable expectations for the revised manuscript. The authors have nicely and strongly argued to my points, in a satisfactorily manner.

Then I acknowledge all the modifications made by the authors who then managed to improve their manuscript.

Reviewer #2 (Remarks to the Author):

The authors have adequately addressed the reviewers' suggestions and I think that the new figure 7 depicting a tentative model of OVATE/SIOFP20 and TRM mode of action is particularly explicative.

I have noticed that some inconsistencies in the discussion of Fig. 3 (page 7 and 8) data are still present. The authors described the sub-cellular localization SITRM17/20a (cytoplasm) and showed the interaction of OVATE and SITRM17/20a by means of BiFC. There is no mention of the co-expression experiments with SITRM17/20a. I suggest to clarify the significance of these data or remove panel D and I from Fig.3.

Reviewer #3 (Remarks to the Author):

The revised manuscript contains a number of changes that substantially improve the clarity of the manuscript. However, the following few issues still need to be addressed:

1- I still think that the title should better reflect the work that is mostly focusing on fruit shape. In their response to the reviewers, the authors did mention many times that they put the main focus on fruit shape without paying much attention to the impact of the genes on other developmental processes. Even for leaf shape, they just described the phenotype but didn't perform any deeper analysis. The work clearly addresses the involvement of OFP and TRM in regulating tomato fruit shape.

2- Page 6, line 130, should refer to Fig.2d rather than to Fig.2b.

3- Page 6-7, line 134 to 137, some statements regarding the Y2H interaction data are not

accurately reflecting the data presented in Figure 2, and should be reformulated. For instance, while mutation of residue D285 in OVATE abolishes the interaction with the TRM proteins tested, mutation of residue D265 in OFP20 doesn't seem to affect the interaction with TRM17/20a. Likewise, mutation in the positively charged R and K amino acid residues of TRMs doesn't seem to have a severe impact in all cases on the interaction with OVATE and OFP20.

4- With regard to in planta subcellular localization experiments, since the re-localization process relies on the interaction between members of the OFP and TRM family, I wonder whether expressing GFP-tagged OVATE and OFP20 proteins in TRM5 KO genetic background, and vice versa with GFP-tagged TRM5 in Ovate and OFP lines would give a clearer indication on the requirement for these interactions. This is easier and more straightforward experiments to perform.

5- The working model presented in Figure 7, may suggest that OVATE and OFP20 have additive/redundant function. That is, a double mutation in the two genes results a more pronounced elongated fruit shape. This might be the case, but direct evidence to draw a conclusion is missing. Could OFP20 complement ovate mutation and conversely is the overexpression of OVATE able to complement sov1 mutation?

Overall the quality of the revised manuscript is significantly improved.

Below, we specifically address the reviewers remaining concerns.

Reviewers' comments:

Reviewer #1 (Remarks to the Author):

Manuscript NCOMMS-18-07950A

"A novel mechanism underlies morphological diversity in plants" by Wu et al.

This manuscript from Wu et al. is a resubmission of a previously reviewed manuscript. It describes a new mechanism underlying the organ shape morphological trait in plants, especially in tomato where the shape of the fruit is determined by the OVATE gene. OVATE belongs to the multigene family encoding the OVATE Family Proteins (OFP). In Arabidopsis and rice, the shapes of a leaf and grain are determined by LONGIFOLIA1 and LONGIFOLIA2, which belongs to the TONNEAU1 Recruiting Motif (TRM) family. In this report, the authors established a link between OFPs and TRMs in the developmental pathway that governs the shape of tomato fruit.

In my previous review I pointed out that the manuscript was somewhat descriptive and that some more information regarding the cellular context would be expected as to go into mechanistic insights about the potential role of these associating proteins in either promoting cell divisions, or influencing in a way the orientation of cell division plates to favor one direction for fruit growth. I acknowledge that this question is rather difficult to address in developing tomato fruits and it may go far beyond the necessary and reasonable expectations for the revised manuscript. The authors have nicely and strongly argued to my points, in a satisfactorily manner.

Then I acknowledge all the modifications made by the authors who then managed to improve their manuscript.

RESPONSE: We are glad to know we have addressed your comments and concerns adequately. And yes we would like to learn how these proteins promote the patterning of cell division and are starting to undertake these are arduous and long term experiments.

Reviewer #2 (Remarks to the Author):

The authors have adequately addressed the reviewers' suggestions and I think that the new figure 7 depicting a tentative model of OVATE/SIOFP20 and TRM mode of action is particularly explicative.

I have noticed that some inconsistencies in the discussion of Fig. 3 (page 7 and 8) data are still present. The authors described the sub-cellular localization SITRM17/20a (cytoplasm) and showed the interaction of OVATE and SITRM17/20a by means of BiFC. There is no mention of the co-expression experiments with SITRM17/20a. I suggest to clarify the significance of these data or remove panel D and I from Fig.3.

RESPONSE: We thank the reviewer for commenting on the figure explaining the model for OFP-TRM interaction and resulting phenotype. As to the second point, the reason to show the interaction of two TRMs in the main figures is to demonstrate that both a MT associated TRM and a cytosolic TRM interact with OVATE. We felt it is important to show the interaction of more than one TRM with OVATE. Otherwise, the readers may have the impression that this interaction is a unique feature of an OFP and one TRM and not a general feature of many OFP (possibly all as we have tested 2 more) and TRMs that carry the M8 motif.

In addition, the tobacco experiments show that the proteins interact by 1. functionally by evaluating the relocalization and disruption of relocalization when we mutate the interacting motifs and 2. mechanically by BiFC. We can only show interaction using this system when the proteins localize to different subcellular structures. There are three major subcellular localizations in the tobacco leaf epidermal cell system namely, cytoskeleton, cytoplasm and nucleus. Thus, if both proteins localize in the cytoplasm, it is difficult to show relocalization upon interaction. This is why we only show for TRM17/20a Y2H with and without mutants and BiFC.

I am not sure if we mentioned this after the first review but the reason we have no Y2H data with TRM3/4 is that in our hands, this TRM does not interact with OVATE in yeast. The proteins clearly interact in the tobacco epidermal leaf cell system. This explains some of the confusion for the reviewer with the earlier version of the manuscript as to why we did certain experiments with certain combinations of OFP and TRM.

Reviewer #3 (Remarks to the Author):

The Responses are after each concern.

The revised manuscript contains a number of changes that substantially improve the clarity of the manuscript. However, the following few issues still need to be addressed:

1- I still think that the title should better reflect the work that is mostly focusing on fruit shape. In their response to the reviewers, the authors did mention many times that they put the main focus on fruit shape without paying much attention to the impact of the genes on other developmental processes. Even for leaf shape, they just described the phenotype but didn't perform any deeper analysis. The work clearly addresses the involvement of OFP and TRM in regulating tomato fruit shape.

RESPONSE: A fruit is an organ and shape could have been studied in depth in any organ of any system. We think that few readers would be interested in a study that only focuses on fruit shape without extensions to other systems. We need to draw in all parties interested in the regulation of organ morphology. The strength of our findings are that OFPs and TRM affect organ shape in many plant species and many organs. We use the tomato fruit as a model because we have a stake in this system and thus exploit the tomato fruit extensively to understand how shape is changed. These seem like trivial experiments, count cells and trace them for size estimates, but they are not because you need to know when during development and where in the organ to count. That would vary for each organ. Natural *ovate* and *sov1* mutant only controls tomato shape and no other organ is affected due, presumably, to redundancy. This is why we did not study in depth the effect of the OFP20 overexpressor on leaves. Moreover, it is well known that overexpression can lead to additional phenotypes that are not controlled by the gene when under its endogenous promoter. Thus, extensive analyses of phenotypes in overexpressors may lead to incorrect conclusions about the function of said gene. The leaf (Arabidopsis)

and grain (rice) shape studies were published and they explain the cellular parameters that influenced the shape in these organs. Basically, the cellular changes corresponding to changes in leaf and grain shape are consistent with our findings. Hence we do not share the reviewer's concern with the title. I think it is clear from the literature and our own data that OFP and TRMs affect organ shape in many different plant species. The tomato study show that organ shape is by physical interaction of the proteins. We can infer that this interaction occurs in the whole plant and in these other systems.

2- Page 6, line 130, should refer to Fig.2d rather than to Fig.2b.

RESPONSE: thanks for catching this.

3- Page 6-7, line 134 to 137, some statements regarding the Y2H interaction data are not accurately reflecting the data presented in Figure 2, and should be reformulated. For instance, while mutation of residue D285 in OVATE abolishes the interaction with the TRM proteins tested, mutation of residue D265 in OFP20 doesn't seem to affect the interaction with TRM17/20a. Likewise, mutation in the positively charged R and K amino acid residues of TRMs doesn't seem to have a severe impact in all cases on the interaction with OVATE and OFP20.

RESPONSE: "In general, mutations in conserved acidic residues in the OFP domain led to reduced or abolished Y2H interaction with different TRMs, whereas mutations in less conserved acidic residues did not change interactions in yeast with SITRM compared to the wild-type allele" is an accurate statement of the data. The Y2H show reduced or abolished interaction when the most conserved acidic residue is changed to the opposite charge in OFP20 and OVATE, respectively. That is an accurate reflection of the data based on the yeast α -galactosidase assay. This assay is more quantitative than plating the yeast on solid media in which case the colonies look less dense. "Conversely, mutations in the basic residue of the SITRM M8 motif reduced the interactions with wild-type OVATE and SIOFP20" is also an accurate statement.

We are not sure what the reviewer is referring to. Unless the reviewer would like us to describe the different combinations and that they have a different effect on the strength of the interaction. However, we don't know what that means nor whether the strength of the interactions varies in vivo in the plants, which is why we keep the discussion about the Y2H to only what is relevant for this study.

4- With regard to in planta subcellular localization experiments, since the re-localization process relies on the interaction between members of the OFP and TRM family, I wonder whether expressing GFP-tagged OVATE and OFP20 proteins in TRM5 KO genetic background, and vice versa with GFP-tagged TRM5 in Ovate and OFP lines would give a clearer indication on the requirement for these interactions. This is easier and more straightforward experiments to perform.

RESPONSE: These experiments would have to be done in tomato and in dividing cells (because that is when shape changes occur). With all due respect, these are by no means easy experiments because we need to study the meaning of the interaction and relocalizations in dividing cells. The tobacco epidermal cells are non dividing. Perhaps the reviewer is of the impression that the dividing cells are on the surface of an organ but they are not for a fruit. Moreover, we don't have sufficiently high expressing OVATE-GFP

plants to see the tag over background and are currently changing the fluorescent tag. It will be a while before those tomato plants are ready to test for expression and then the rest. Lastly, these experiments are also not trivial as the timing of shape changes is difficult to ascertain in the developing ovary. Regardless, it will be at least two years before we can ask the question the reviewer is referring to. The experiments presented in this manuscript have taken us years to accomplish with limited specific funding for the work. I think we present a strong case of a novel interaction underlying organ shape in all plants and all organs with the data we have.

5- The working model presented in Figure 7, may suggest that OVATE and OFP20 have additive/redundant function. That is, a double mutation in the two genes results a more pronounced elongated fruit shape. This might be the case, but direct evidence to draw a conclusion is missing. Could OFP20 complement ovate mutation and conversely is the overexpression of OVATE able to complement sov1 mutation?

RESPONSE: the reviewer is correct that OFP20 and OVATE may function redundantly. However, that was not the impression we wanted to give because we have no evidence that they are redundant *in vivo*. We like to think that OVATE sets up the pattern (ie slight change in cell division very early in development) and OFP20 extends the change in pattern (somewhat later in development). Based on expression patterns of these two OFPs that is a reasonable assumption albeit as speculative as the complementing roles of the two genes. To avoid confusion, we have removed figure 7b primarily upon the suggestion of the editor as well as your comment.

Overall the quality of the revised manuscript is significantly improved.

RESPONSE: Thank you.